# Scaling Multimodal Theory-of-Mind with Weak-to-Strong Bayesian Reasoning

## Abstract

Theory-of-Mind (ToM) enables individuals to the mental states of the others, such as thoughts, beliefs, and desires. To replicate this cognitive ability in machines, especially under complex multimodal environments, recent advances combine Bayesian-based state inference with deep learning models to estimate mental states, where the Bayesian model handles state transitions and a language model (LM) estimates the likelihood of intermediate states. However, while post-training an LM to specialise in ToM tasks improves performance, the computational cost increases as the LM scales, limiting the model size to 7 billion parameters. Despite this post-training process, smaller LMs still struggle with the physical and mental modelling demands of ToM due to their limited world knowledge and reasoning capacity. To address this, we propose a scalable solution that leverages the strengths of larger LMs (up to 70 and 405 billion parameters, respectively), including their vast world knowledge and atomic-level reasoning capabilities, without increasing post-training resource requirements. Our method transfers ToM-specific behaviours from a post-trained small LM to guide the latent reasoning of a larger LM during test time. This *weak-to-strong* control mechanism enables the larger LM to improve Bayesian likelihood estimation at each inference step, harnessing its reasoning power in ToM scenarios while reducing the need for additional training resources. Extensive experiments demonstrate the significant effectiveness of our scaled approach. It is better at inferring human mental states in complex and interactive environments, outperforming the state-of-the-art solution by $\sim 4.6\%$ across multiple tasks on the multimodal ToM benchmark and unseen scenarios.

## 1 Introduction

A key aspect of human social cognition is Theory-of-Mind (ToM)—the ability to comprehend and attribute mental states such as beliefs, desires, and intentions to ourselves and others. This capacity allows individuals to recognize that others may have perspectives and motivations distinct from their own, forming the foundation of social understanding and interaction (Dennett, 1988; Gopnik & Wellman, 2012). Building on this concept, a critical challenge lies in enabling artificial intelligence (AI) to acquire human-level ToM capabilities. Equipping AI systems with such abilities could significantly enhance their potential for human-like interactions and unlock broader capacities for commonsense and context-aware reasoning (Lake et al., 2017; Wu et al., 2021; Ma et al., 2023).

Among the ongoing efforts to advance ToM abilities in machines, two broadly classed strategies are the highly structured Bayesian approaches and the less structured end-to-end approaches, respectively: *(i)* Bayesian ToM models use cognitively structured probabilistic frameworks to represent causal relationships between the mind and the world, enabling inverse inference of mental states from sparse behavioral observations (Baker et al., 2017; Jara-Ettinger, 2019; Shu et al., 2021). These models are highly interpretable and support key ToM functions like explanation and inductive learning, excelling in well-defined domains with precise predictions and few-shot generalization (Shum et al., 2019; Zhi-Xuan et al., 2022). However, they require extensive inductive constraints (e.g., abstractions, priors, and causal relationships) provided by experts, **limiting their scalability to complex and unconstrained practical environments.** *(ii)* End-to-end models of ToM involve training deep models directly on ToM data (Rabinowitz et al., 2018; Shu et al., 2021; Sclar et al., 2022). These models, such as ToMnet (Rabinowitz et al., 2018), autonomously learn complex patterns and relationships from data without explicitly concepts like agents, beliefs, and goals. However, because these models

do not explicitly encode principles of physics or theories of psychology, they lack transparency and may draw conclusions that violate physical laws, logical coherence, or commonsense reasoning. Additionally, their data-driven nature makes them **less reliable in, and adaptable to, dynamic or novel environments**, particularly where data is sparse or unrepresentative (Sap et al., 2022; Zhi-Xuan et al., 2022; Ullman, 2023; Strachan et al., 2024).

The recent interdisciplinary milestone, BIPALM (Bayesian inverse planning accelerated by language model), merges the strengths of Bayesian ToM models and deep learning to facilitate robust reasoning in complex multimodal scenarios (Jin et al., 2024). It uses a Bayesian ToM model to predict agents' mental states in multimodal scenarios, with a language model (LM) estimating the probabilities of these Bayesian intermediate states based on multimodal inputs. For accurate Bayesian ToM inference, the LM undergoes an additional post-training process tailored to the target multimodal contexts. However, this reliance on post-training significantly increases computational demands and limits the scalability of likelihood models, restricting the size of LMs to around seven billion parameters. ToM intricately intertwines **open-domain world knowledge** and **implicit reasoning** to ground human mental states within their corresponding physical environments. As environments and their associated queries evolve, reliance on smaller post-trained LMs introduces a critical trade-off: while smaller LMs excel in adapting to specific ToM tasks within particular environments, larger LMs are indispensable for harnessing broader world knowledge and advanced reasoning capabilities. **This inherent tension poses a significant challenge, requiring us to improve the scalability of Bayesian ToM methods in handling dynamically evolving, multimodal ToM scenarios**

**Motivated by the scalability challenge of current Bayesian ToM methods**, we propose a scalable Bayesian inference solution that generalises to complex and dynamic environments (Tab.5 in App.A compares our technical contributions with other solutions). Our approach introduces a *weak-to-strong* control mechanism where post-trained smaller LMs specialise in ToM-specific tasks by capturing likelihood inference patterns during Bayesian ToM reasoning. These ToM behaviours are then transferred to larger LMs during test time, aligning the larger models' reasoning trajectories with the structured requirements of Bayesian inverse planning. In this framework, the larger LM acts as the *primary policy model*, leveraging its extensive world knowledge and reasoning capabilities. Importantly, the reasoning trajectory of the larger LM is structured by the Bayesian framework, ensuring consistency, robustness, and interpretability. This design avoids additional post-training costs for larger LMs while enabling scalable use of large models up to **70B** or **405B** parameters.

In particular, we focus on the pattern shifts observed in smaller LMs before and after ToM post-training, treating these shifts as *ToM behaviours* that guide larger LMs. This ToM-behaviour transfer ensures that larger LMs, redirected as **aligned policy models**, follow reasoning trajectories primarily inferenced by the Bayesian framework. The experiments demonstrate that this weak-to-strong control mechanism significantly enhances the generalizability of Bayesian inference, achieving a $\sim 4.6\%$ improvement in accuracy over state-of-the-art methods, even in dynamic and unseen environments.

## 2 METHODOLOGY: SCALED BAYESIAN REASONING ON MULTIMODAL TOM

Our scaled Bayesian reasoning infers an agent's mental state based on video and text inputs. While the extension of Bayesian methods to deep models can reverse engineer human ToM reasoning in multiple domains, it also includes LMs for multimodal inputs: (1) building unified representations about a scene, a person's actions, and the mental state hypotheses from multimodal inputs, and (2) post-training an LM to conduct contextual inverse symbolic planning, based on unified symbolic representations (Jin et al., 2024). Then, as shown in Fig.1, the LM used in our scaled Bayesian reasoning is from 7B up to 405B parameters at test-time compute, avoiding additional post-training.

### 2.1 DATA REPRESENTATION

Instead of single-modality input, this study integrates both visual and textual data into a unified symbolic representation, enabling a more comprehensive understanding of the context and human behaviour: *(i)* One visual perception module converts visual data into symbolic representations by using the method from Blukis et al. (2022) to generate a voxel map and construct a scene graph for each video frame. *(ii)* For text parsing, textual information is extracted into symbolic representations of the initial state and subsequent actions. The parser processes the text by breaking it down into

Figure 1: *(left)* The large LM operates as a scaled **policy model** to estimate the likelihood of an agent's actions in dynamic environments, based on multimodal symbolic inputs (video and description). *(right)* The latent reasoning of the large LM is guided by the ToM behaviours from post-trained small LMs, which acts as a weak-to-strong control. Overall, Bayesian inverse planning compares hypotheses about the agent's goal and belief, using the large LM as a policy model to infer ToM.

components: the state of the environment, the actions taken by humans, and the question. It translates these into natural language descriptions, such as *"the pear is inside the basket"* for the environment's state, *"walks toward the kitchen"* for action commands, and two potential goals like *"to retrieve the pear"* and beliefs such as *"the pear is inside the basket"* or *"the pear is not inside the basket."* *(iii)* finally, in unifying multimodal information, the fusion step aligns and integrates input stream information by converting scene graphs from video into predicates that describe spatial relationships and object statuses, analogous to text-derived predicates. We create a symbolic representation of the initial state by combining predicates from both video and text, aligning and updating state predicates with new video frame information at each time step, and constructing a symbolic state and action sequence. This process begins from the initial state, with actions parsed from the text aligned with video-detected actions and divided into intervals corresponding to each action. See App.B.2 for more details on data preprocessing.

## 2.2 INFERRING HUMAN MENTAL STATES

To infer human mental states, we generate and evaluate hypotheses about human intentions by employing Bayesian inverse planning. These hypotheses are assessed through action likelihoods, which directly inform the posterior probabilities of different mental state explanations. To compute these crucial likelihoods, an LM is controlled by our weak-to-strong mechanism to serve as the policy model. This approach harnesses the world knowledge and reasoning capabilities of large LLMs while aligning their behaviour patterns to ToM tasks without extensive computational requirements.

**Behaviour modelling: a Markov decision process formulation** The behaviour of an agent can be formulated as a forward generative model based on a Partially Observable Markov Decision Process (POMDP), defined by the tuple $\langle S, A, \mathcal{T}, G, R, \Omega, O, \gamma \rangle$ (Kaelbling et al., 1998; Jin et al., 2024). Here, $s^t \in S$ and $a^t \in A$ represent the state and action at time $t$, respectively. $\mathcal{T}(s^t|s, a)$ denotes the state transition probabilities. The goal $g \in G$ determines the reward $r^t = R(s^t, a^t, g)$. The agent's observation $o^t \in \Omega$ is obtained via the observation function $o^t = O(s^t)$. The discount factor is $\gamma \in (0, 1]$. Crucially, the agent's belief, $b(s)$, is a probability distribution over the state. This belief is dynamically updated during belief evolution $P(b^\tau \mid b^{\tau-1}, s^\tau)$, where $b(s)$ is factorized into probabilities over the possible locations of individual objects.

**Inverse inference: from observed behaviours to the mental states** While the POMDP formulates a forward model of agent behaviour, the heart of this Bayesian approach is to **invert** this process – inferring the agent's goals and beliefs from observed behaviours, i.e., actions (Baker et al., 2017). Assuming deterministic state transitions for simplicity, we jointly infer the agent's goal and belief based on observed states and actions. The posterior probability of an agent's goal $g$ and belief $b^t$ given a sequence of observed states $s^{1:t}$ and actions $a^{1:t-1}$ is expressed as:

$$P(g, b^t|s^{1:t}, a^{1:t-1}) \propto \prod_{\tau=1}^{t} \pi(a^\tau|g, b^\tau) P(b^\tau|b^{\tau-1}, s^\tau) P(b^0) P(g), \tag{1}$$

where $\pi(a^\tau|g, b^\tau)$ represents the agent's policy, which captures the probability of taking action $a^\tau$ given a goal $g$ and belief $b^\tau$. This reflects the agent's decision-making process, influenced by its goals and the current state of its beliefs. Belief evolution is modeled as $P(b^\tau|b^{\tau-1}, s^\tau) =$

$\frac{P(s^\tau|b^{\tau-1})P(b^{\tau-1})}{P(s^\tau)}$, where $P(s^\tau|b^{\tau-1})$ is the likelihood of observing $s^\tau$ given the prior belief $b^{\tau-1}$, and $P(b^{\tau-1})$ represents the prior belief at the previous step. This belief evolution describes how the belief $b^\tau$ evolves from the prior belief $b^{\tau-1}$ after observing a new state $s^\tau$, and it follows Bayesian principles, ensuring that new evidence incrementally refines the agent's understanding of its environment. For example, if an object is observed inside a container at time $\tau$, the corresponding belief about the object's location is updated accordingly. **In practice,** $\pi(a^\tau|g, b^\tau)$ and $P(b^\tau|b^{\tau-1}, s^\tau)$ can be approximated using likelihood generated by a language model. $P(b^0)$ is set as a uniform distribution to reflect equal uncertainty about all possible object locations at the start, while $P(g)$ encodes prior knowledge about the likelihood of different goals.

To compare different hypotheses about the agent's goals and beliefs, we evaluate their relative log-likelihoods from a given set of hypotheses, separating the contributions from the current time step and the accumulated effect of prior steps. Consider two hypotheses, $H_1 = \langle g_1, b_1^t \rangle$ and $H_2 = \langle g_2, b_2^t \rangle$, representing different goal-belief pairs. Their relative log-likelihoods are compared as:

$$\log \frac{P(g_1, b_1^t|s^{1:t}, a^{1:t})}{P(g_2, b_2^t|s^{1:t}, a^{1:t})} = \underbrace{\log \frac{\pi(a^t|g_1, b_1^t)}{\pi(a^t|g_2, b_2^t)} + \log \frac{P(b_1^t|\hat{b}^{t-1}, s^t)}{P(b_2^t|\hat{b}^{t-1}, s^t)}}_{\text{Current step comparison}} + \underbrace{\sum_{\tau=1}^{t-1} \log \frac{\pi(a^\tau|g_1, \hat{b}^\tau)}{\pi(a^\tau|g_2, \hat{b}^\tau)}}_{\text{Prior steps comparison}}. \quad (2)$$

The first term compares log-likelihoods of actions and belief updates at the current step, reflecting how each hypothesis explains the agent's latest behavior. The second term sums log-likelihoods from prior steps, ensuring the entire action history informs the hypothesis evaluation. **In practice**, the belief $\hat{b}^\tau$ is ***not explicitly*** updated as a posterior distribution $P(b^\tau \mid b^{\tau-1}, s^\tau)$. Instead, it is symbolically approximated as a structured hypothesis (e.g., possible object locations) that represents the agent's understanding of the environment at each step. By comparing action likelihoods $\pi(a^\tau \mid g, s^\tau, \hat{b}^\tau)$ across hypotheses, beliefs are implicitly evaluated and updated.

**Weak-to-strong controlled large policy model**   When augmenting likelihood estimation with the guided large LM's broad generalization capabilities, we scale up the LM used only for test-time computing and avoid the direct post-training on large LM. The true policy $\pi(a^\tau \mid g, b^\tau)$ is estimated through a language model ($\pi$)-estimated probability $\tilde{\pi}(a^t \mid s^t, g, \hat{b}^t)$:

$$\pi(a^\tau \mid g, b^\tau) = \tilde{\pi}(a^t \mid s^t, g, \hat{b}^t) + \varepsilon, \quad (3)$$

where $\varepsilon$ represents the inherent approximation error. When applied to the Bayesian inverse planning framework equation 1, the posterior probability is expressed as:

$$P(g, b^t \mid s^{1:t}, a^{1:t-1}) \propto \prod_{\tau=1}^{t} [\tilde{\pi}(a^\tau \mid s^\tau, g, \hat{b}^\tau) + \varepsilon] \cdot P(b^\tau \mid b^{\tau-1}, s^\tau) \cdot P(b^0)P(g). \quad (4)$$

It integrates the approximation into Bayesian reasoning, considering $s_i, b_i, g_i, a_i$ updates over time.

POST-TRAINING STAGE: ToM OPTIMIZATION   To allow direct use of the LM-estimated probability, we aim to reduce $\varepsilon$ via a post-training stage to align LM's pretrained capability to current target situations. The initial phase of reducing $\varepsilon$ involves refining a scenario-specific post-training policy $\pi^{\mathcal{E}}$ on an action-policy experience pool. This pool $\mathcal{D}$ is defined as: $\mathcal{D} = \{(s_i, b_i, g_i, a_i)\}_{i=1}^{N}$, where $s_i, b_i, g_i, a_i$, and $N$ denote sequences of states, beliefs, goals, actions, the number of data points sourced from multimodal situations. The objective function guiding post-training is:

$$\mathcal{L}(\pi^{\mathcal{E}}) = -\sum_{i=1}^{N} \log \pi^{\mathcal{E}}(a_i \mid s_i, b_i, g_i). \quad (5)$$

Here, $\pi^{\mathcal{E}}$ learns the human ToM behaviour patterns effectively, allowing the language model policy to adeptly learn and respond to complex ToM environments. Due to computational constraints, this post-training process is typically applied to smaller models.

INFERENCE STAGE: LARGE POLICY MODEL WITH BEHAVIORAL GUIDANCE   During test inference, we leverage the behaviour acquired by the post-trained smaller LM to guide the reasoning of a larger,

more capable LM. This approach dynamically adjusts the output of the large LM based on the shift observed between a post-trained small LM $\pi^{\mathcal{E}}$ and a naive small LM $\pi^{\mathcal{N}}$. At each inference step $t$, the overall policy distribution for the redirected large model is given by:

$$\bar{\pi}(a^t \mid s^t, g, \hat{b}^t) = \frac{1}{\bar{Z}} \pi^{\mathcal{L}}(a^t \mid s^t, g, \hat{b}^t) \frac{\pi^{\mathcal{E}}(a^t \mid s^t, g, \hat{b}^t)}{\pi^{\mathcal{N}}(a^t \mid s^t, g, \hat{b}^t)}, \tag{6}$$

where $\pi^{\mathcal{L}}(a^t \mid s^t, g, \hat{b}^t)$ represents the policy distribution from the naive large LM. The post-training effect to policy function is *approximated* through the ratio $\frac{\pi^{\mathcal{E}}(a^t \mid s^t, g, \hat{b}^t)}{\pi^{\mathcal{N}}(a^t \mid s^t, g, \hat{b}^t)}$, offering an on-the-fly redirecting mechanism. The normalization factor is calculated by $\bar{Z} = \sum_{a^t} \pi^{\mathcal{L}}(a^t \mid s^t, g, \hat{b}^t) \frac{\pi^{\mathcal{E}}(a^t \mid s^t, g, \hat{b}^t)}{\pi^{\mathcal{N}}(a^t \mid s^t, g, \hat{b}^t)}$. It ensures that the resulting probabilities remain a valid distribution, reflecting both the post-training adjustments and the foundational likelihood from the larger model. Our overall method facilitates ToM behaviour transfer from the post-trained small LM ($\pi^{\mathcal{E}}$) to the larger LM ($\pi^{\mathcal{L}}$), scaling the capabilities of the policy model in Bayesian inference at test-time. The Theroem 1 and its proof in the appendix C are provided for theoretical support.

## 3 EXPERIMENTS

We examine the **scaling benefits** of this Bayesian weak-to-strong reasoning in multimodal ToM tasks. For the *strong* component, we scale up the large LMs to 70B and 405B parameters. In contrast, for the *weak* component, we reduce the size of the small LMs from 8B to 4B parameters. First, the results reveal a positive correlation between model size and ToM capabilities, especially when the larger models are guided by the post-trained behaviours of the smaller models. Interestingly, these post-trained behaviours are also effectively captured by smaller LMs. We also illustrate how the large LMs are progressively redirected to the answer space during the Bayesian process.

### 3.1 SETUP

**Datasets**    *(i) For post-training*, we use MMToM sampled from an apartment environment simulator, Virtual Home (Puig et al., 2018), using the procedural methods described by Jin et al. (2024). The dataset comprises 1,000 procedurally synthesized videos within a realistic household simulator, each annotated with ground-truth labels for states, goals, beliefs, and actions. These precise annotations are used to train our conditional prediction model for action $a_i$ given state $s_i$, belief $b_i$, and goal $g_i$: $P(a_i \mid s_i, b_i, g_i)$. By leveraging this synthetic dataset, the inverse symbolic planner—comprising a large base LM guided by a smaller, post-trained LM—acquires robust exposure to diverse scenarios. *(ii) For evaluation*, we use the MMToM-QA (Jin et al., 2024), an evaluation benchmark aimed at evaluating ToM reasoning over multimodal situations. The dataset consists of 134 videos, each showing a person searching for household objects, with an average of 1,462 frames per video representing approximately 36 human actions. These videos are accompanied by 600 questions (detailed in appendix §D.1), evenly divided between the categories of belief inference (with 1.1, 1.2, and 1.3 subtasks) and goal inference (with 2.1, 2.2, 2.3, and 2.4 subtasks). Each question is paired with a video clip and a detailed textual description The questions are designed to assess the ability of models to infer goals and beliefs jointly, providing a richer assessment of multimodal ToM capabilities. It supports three setups, including multimodal, text-only, and video-only inputs.

**Baselines**    We include three types of baselines in our evaluation to benchmark our model's performance on the MMToM-QA dataset. *For text-only evaluation*, we compare performance in the text-only subset of MMToM-QA using various large language models (LLMs), including GPT-4 (OpenAI, 2023a), GPT-3.5, GPT-J-6B (Wang & Komatsuzaki, 2021), and Llama-2-7B (Touvron et al., 2023). Advanced prompting methods, such as SimToM (Wilf et al., 2024) and SymbolicToM (Sclar et al., 2023), which enhance GPT-4's reasoning capabilities, provide additional baselines (e.g., SimToM with GPT-4 and SymbolicToM with GPT-4). *For multimodal evaluation*, we include GPT-4V (OpenAI, 2023a), InstructBLIP (Dai et al., 2023), Video-Llama-2 (Zhang et al., 2023), and LLaVA (Liu et al., 2023), BIPALM (Jin et al., 2024). These methods employ sampled frames from each video to evaluate their proficiency in multimodal ToM reasoning. *For human*, 180 participants answer 120 randomly sampled questions, covering all question types, as reported by Jin et al. (2024).

Table 1: Comparisons between humans and models across task types from 1.1 to 2.4 are provided. The best results for each modal setting are highlighted in **bold**. The second best results in multimodality are underlined. Rows of ours are highlighted in color.

| | method | belief inference | | | | goal inference | | | | | all |
| | | 1.1 | 1.2 | 1.3 | avg. | 2.1 | 2.2 | 2.3 | 2.4 | avg. | |
|---|---|---|---|---|---|---|---|---|---|---|---|
| text only | *Human* | 96.0 | 95.8 | 81.3 | 91.0 | 85.8 | 76.7 | 65.0 | 68.3 | 74.0 | 82.5 |
| | GPT-4 | 97.0 | 12.0 | 77.0 | 62.0 | 48.0 | 42.7 | 2.7 | 42.7 | 34.0 | 48.0 |
| | GPT-3.5 | 81.0 | 11.0 | 39.0 | 43.7 | 46.7 | 16.0 | 21.3 | 48.0 | 33.0 | 38.3 |
| | GPT-J-6B | 56.0 | 53.0 | 38.0 | 49.0 | 52.0 | 50.7 | **50.7** | 56.0 | 52.3 | 50.7 |
| | Llama-2-7B | 64.0 | 55.0 | 50.0 | 56.3 | 49.3 | 48.0 | 41.3 | 38.7 | 44.3 | 50.3 |
| | SimToM *w/* GPT-4 | 96.0 | 15.0 | 82.0 | 64.3 | 61.3 | 44.0 | 2.7 | 54.7 | 40.7 | 52.5 |
| | SymbolicToM *w/* GPT-4 | **100** | 61.0 | 74.0 | 78.3 | 73.3 | 66.7 | 0.0 | 50.7 | 47.7 | 63.0 |
| | BIPALM *w/* GPT-J-6B | 88.0 | **69.0** | 88.0 | 81.7 | **77.3** | **68.0** | 30.7 | **70.7** | **61.7** | **71.7** |
| | BIPALM *w/* Llama-2-7B | 89.0 | 68.0 | **90.0** | **82.3** | 54.7 | 66.7 | **50.7** | 62.7 | 58.7 | 70.5 |
| video only | *Human* | 69.1 | 64.3 | 86.4 | 73.3 | 58.5 | 60.0 | 76.7 | 63.3 | 64.6 | 68.9 |
| | InstructBLIP-13B | 56.0 | 50.0 | 42.0 | 49.3 | 56.0 | 45.3 | 54.7 | 53.3 | 52.3 | 50.8 |
| | Video-Llama-2-13B | 24.0 | 32.0 | 67.0 | 41.0 | 50.7 | 45.3 | **56.0** | 52.0 | 51.0 | 46.0 |
| | LLaVA-7B | 33.0 | 15.0 | 69.0 | 39.0 | 44.0 | 24.0 | **56.0** | 57.3 | 45.3 | 42.2 |
| | GPT-4V | 64.0 | 34.0 | 39.0 | 45.7 | 54.7 | 26.7 | 48.0 | 56.0 | 46.3 | 46.0 |
| | BIPALM *w/* GPT-J-6B | 63.0 | 57.0 | **72.0** | **64.0** | 45.3 | **62.7** | 50.7 | **62.7** | 55.3 | 59.7 |
| | BIPALM *w/* Llama-2-7B | **69.0** | **63.0** | 60.0 | **64.0** | **62.7** | 54.7 | 53.3 | **62.7** | **58.3** | **61.2** |
| multimodal | *Human* | 95.8 | 96.7 | 100 | 97.5 | 90.0 | 91.7 | 83.3 | 88.9 | 88.5 | 93.0 |
| | InstructBLIP-13B | 62.0 | 52.0 | 32.0 | 48.7 | 46.7 | 29.3 | 42.7 | 60.0 | 44.7 | 46.7 |
| | Video-Llama-2-13B | 36.0 | 38.0 | 52.0 | 42.0 | 36.0 | 41.3 | 30.7 | 45.3 | 38.3 | 40.2 |
| | LLaVA-7B | 46.0 | 14.0 | 69.0 | 43.0 | 65.3 | 22.7 | 40.0 | 48.0 | 44.0 | 43.5 |
| | GPT-4V | **94.0** | 13.0 | 59.0 | 55.3 | 56.0 | 26.7 | 4.0 | 52.0 | 34.7 | 44.0 |
| | BIPALM *w/* GPT-J-6B | 90.0 | 69.0 | 86.0 | 81.7 | 68.0 | 78.7 | 56.0 | 73.3 | 69.0 | 75.3 |
| | BIPALM *w/* Llama-2-7B | 88.0 | 68.0 | 85.0 | 80.3 | 62.7 | 77.3 | 72.0 | **80.0** | 73.3 | 76.7 |
| | **Ours (*w/* Llama-3.1-405B)** | 92.0 | **76.0** | **93.0** | **87.0** | 73.3 | **80.0** | **76.0** | 78.7 | **77.0** | **81.3** |

**Post-training**   We post-train Llama (Touvron et al., 2023; Dubey et al., 2024) as a policy model in our Bayesian framework with LoRA (Hu et al., 2022), as outlined in Tab.6. Following the setup recommended by Jin et al. (2024), we use a learning rate of 1e-3 over 3 epochs. LoRA is configured with a rank of 16 and an alpha value of 32 for the 7B and 8B LMs. For 70B, we use a lower rank of 8 and an alpha of 16. They are carefully tuned to optimize performance across varying LM sizes.

## 3.2   MAIN RESULTS

Tab.1 uses human performance as the gold standard. Humans clearly outperform all models (with 93.0% accuracy) when provided with multimodal input. This result highlights the **critical role of integrating both visual and textual modalities** in achieving an immersive and context-aware perception of the ToM situation. Accordingly, our ToM method also incorporates multimodal inputs. In *belief inference, which is strongly linked to world knowledge*, models like GPT-4 and GPT-3.5 perform exceptionally well, particularly on task 1.1, where GPT-4 achieves an accuracy of 94%. This result underscores the importance of large-scale models in capturing and applying vast amounts of pretrained world knowledge. However, despite their impressive performance in belief inference, these models do not perform as effectively on *goal inference, where adaptation to specific ToM contexts and dynamic environments is crucial*. **This highlights the need for models to be better aligned with the specific requirements of ToM scenarios.** Models with smaller scales, such as those with 6B, 7B, and 13B parameters, face inherent capability limitations, which restrict their performance on belief inference tasks, particularly when compared to larger models like GPT-4 on task 1.1. However, these smaller models, such as BIPALM w/ GPT-J-6B and Llama-2-7B, benefit from post-training specifically designed for ToM scenarios. This allows them to perform better on goal inference tasks, where understanding and adapting to scenario-specific environmental dynamics is essential. **Despite their size constraints, these models demonstrate the value of targeted post-training in compensating for the lack of large-scale pretrained knowledge.** Our approach goes beyond seesaw effects in prior methods and has both strengths: while its strong component leverages the extensive world knowledge embedded in large pretrained models, also its weak component incorporates post-

Table 2: Scaling-up performance on strong component (large LMs) in weak-to-strong control.

| LM | config | belief inference | | | | goal inference | | | | | all |
|---|---|---|---|---|---|---|---|---|---|---|---|
| | | 1.1 | 1.2 | 1.3 | avg. | 2.1 | 2.2 | 2.3 | 2.4 | avg. | |
| Llama-2 | 7B-zero-shot | 44.00 | 37.00 | 84.00 | 55.00 | 64.00 | 65.33 | 62.67 | 64.00 | 64.00 | 60.14 |
| | 7B-post-trained | 80.00 | 60.00 | 89.00 | 76.33 | 74.67 | 60.00 | **78.67** | 66.67 | 70.00 | 72.71 |
| | 70B-zero-shot | 64.00 | 47.00 | **93.00** | 68.00 | 56.00 | 72.00 | 25.33 | 70.67 | 56.00 | 61.14 |
| | 70B-post-trained | **90.00** | **70.00** | 87.00 | 82.33 | **78.67** | **76.00** | 61.33 | 72.00 | 72.00 | 76.43 |
| | **70B-ours** | 89.00 | **70.00** | 90.00 | **83.00** | 73.33 | 74.67 | 76.00 | **73.33** | **74.33** | **78.05** |
| Llama-3 | 8B-zero-shot | 88.00 | 72.00 | 91.00 | 83.67 | 65.33 | 57.33 | 13.33 | 53.33 | 47.33 | 62.90 |
| | 8B-post-trained | **92.00** | 72.00 | 83.00 | 82.33 | **77.33** | 73.33 | 72.00 | 70.67 | 73.33 | 77.19 |
| | 70B-zero-shot | 69.00 | 67.00 | **95.00** | 77.00 | 42.67 | 70.67 | 16.00 | 52.00 | 45.33 | 58.90 |
| | 70B-post-trained | 91.00 | 70.00 | 89.00 | 83.33 | 73.33 | **74.67** | 44.00 | 69.33 | 65.33 | 73.05 |
| | **70B-ours** | 91.00 | **75.00** | 92.00 | **86.00** | 68.00 | 72.00 | **74.67** | **78.67** | 73.33 | **78.76** |
| Llama-3.1 | 8B-zero-shot | 88.00 | 72.00 | 91.00 | 83.67 | 65.33 | 62.67 | 22.67 | 54.67 | 51.33 | 65.19 |
| | 8B-post-trained | 90.00 | 71.00 | 93.00 | 84.67 | 69.33 | 72.00 | 62.67 | 72.00 | 69.00 | 75.71 |
| | 70B-zero-shot | 85.00 | 63.00 | 93.00 | 80.33 | 72.00 | 76.00 | 16.00 | 61.33 | 56.33 | 66.62 |
| | 70B-post-trained | 91.00 | 69.00 | **95.00** | 85.00 | 69.33 | 80.00 | 29.33 | 69.33 | 62.00 | 71.86 |
| | 405B-zero-shot | 86.00 | 70.00 | 90.00 | 82.00 | 73.33 | 78.67 | 21.33 | 66.67 | 60.00 | 69.43 |
| | **70B-ours** | 90.00 | 74.00 | 93.00 | 85.67 | **74.67** | 77.33 | 70.67 | 76.00 | 74.67 | 79.38 |
| | **405B-ours** | **92.00** | **76.00** | 93.00 | **87.00** | 73.33 | **80.00** | **76.00** | **78.67** | **77.00** | **81.29** |

training to the ToM contexts and environmental dynamics required. This dual advantage allows a balanced performance across both task types (belief & goal inference), **with an overall 81.3% accuracy on multimodal tasks and exhibits a 4.6% improvement over the existing best baseline.**

### 3.3 STRONGER LARGE LMS ENHANCE LIKELIHOOD ESTIMATION IN BAYESIAN INFERENCE

In the Bayesian framework, we explore the role of LMs in likelihood estimation and examine how their scale and post-training affect performance across various ToM tasks. According to Tab.2, *(i)* **our results demonstrate a positive correlation between LM size and ToM task performance.** For instance, in the zero-shot setting of Llama-3.1, the 405B model achieves an accuracy of 69.43%, outperforming both the 8B model (65.19%) and the 70B model (66.62%). Notably, the performance of the 405B model approaches that of the post-trained Llama2-7B. Furthermore, the improvement from 70B to 405B suggests that the benefits of scaling have not yet reached saturation, indicating potential for further gains with larger models. *(ii)* **Post-training significantly enhances LMs' performance on ToM tasks, even when Larger LMs already perform well in zero-shot scenarios.** This effect is consistent across model sizes, from smaller models such as 7B/8B to larger models up to 70B, regardless of the specific version (Llama-2, Llama-3, or Llama-3.1). For belief inference tasks, which are closely tied to world knowledge, post-training helps align the large models' knowledge more precisely with the input questions. For goal inference tasks, which are linked to environmental dynamics, post-training refines the models' atomic-level reasoning (i.e., predicting $a$ based on $s, b, g$), resulting in greater improvements compared to belief inference. This suggests that post-training provides a more substantial benefit for tasks that require dynamic reasoning. *(iii)* **Our weak-to-strong control approach** *approximates* **the benefits of direct post-training in Bayesian inference.** When comparing models such as Llama-2, Llama-3, and Llama-3.1, we find that direct post-training on the 70B model, even with adjusted hyperparameters from the 8B model (e.g., reducing the alpha value from 32 to 16), does not produce results as robust as our method. We attribute this to the difficulty of finding optimal hyperparameters for larger models, which require more extensive tuning. In contrast, our weak-to-strong control, which uses a well-trained smaller LM to guide the larger LMs, allows for more consistent improvements without the need for extensive hyperparameter trials.

### 3.4 DOWNSIZED SMALL LMS KEEP EFFECTIVENESS IN WEAK-TO-STRONG CONTROL

In the Bayesian framework, prior experiments show that post-trained behaviours from small LMs can effectively guide the pretrained capabilities of larger LMs during test time. To further study the role of post-training to weak-to-strong control, Tab.3 investigates whether post-trained behaviours can be learned effectively with reduced computational resources, while the pretrained capabilities of larger LMs are still available. Specifically, we examine whether *downsized* smaller LMs can effectively

Table 3: Scaling-down effect on weak part (small LMs) in weak-to-strong controlled Bayesian reasoning. All models are based on Llama3.1. Rows of our method are highlighted in color.

| LM | config | belief inference | | | | goal inference | | | | | all |
|---|---|---|---|---|---|---|---|---|---|---|---|
| | | 1.1 | 1.2 | 1.3 | avg. | 2.1 | 2.2 | 2.3 | 2.4 | avg. | |
| 8B | zero-shot | 88.00 | 72.00 | 91.00 | 83.67 | 65.33 | 62.67 | 22.67 | 54.67 | 51.33 | 65.19 |
| | post-trained | 90.00 | 71.00 | 93.00 | 84.67 | 69.33 | 72.00 | 62.67 | 72.00 | 69.00 | 75.71 |
| | **8B ⤳ 70B** | **90.00** | **74.00** | **93.00** | **85.67** | **74.67** | **77.33** | **70.67** | **76.00** | **74.67** | **79.38** |
| 4B wid. | zero-shot | 79.00 | 69.00 | 89.00 | 79.00 | 60.00 | 69.33 | 24.00 | 52.00 | 51.33 | 63.19 |
| | post-trained | 90.00 | 72.00 | 87.00 | 83.00 | 70.67 | 72.00 | 68.00 | **78.67** | 72.33 | 76.90 |
| | **4B-width ⤳ 70B** | **90.00** | 71.00 | **90.00** | 83.67 | **74.67** | 74.67 | 76.00 | 73.33 | **74.67** | 78.52 |
| 4B dep. | zero-shot | 91.00 | **74.00** | 88.00 | 84.33 | 69.33 | **77.33** | 20.00 | 66.67 | 58.33 | 69.48 |
| | post-trained | 91.00 | 71.00 | 90.00 | 84.00 | 65.33 | 65.33 | 76.00 | **69.33** | 69.00 | 75.43 |
| | **4B-depth ⤳ 70B** | **91.00** | 72.00 | **91.00** | **84.67** | 72.00 | 74.67 | **84.00** | 64.00 | **73.67** | 78.38 |

Table 4: Transfer performance of the Bayesian method with different scaling settings (zero-shot, direct post-training, and our weak-to-strong control) from the apartment scenario to various unseen environments. All models are based on Llama3.1. Results are average accuracy of *belief inference/goal inference/overall* for each scenario. Detailed unseen scenarios and results are in §D.8&D.9.

| | solution | apartment *(seen)* | Andersen tales | ancient Egyptian | outer space | wild west | medieval castle |
|---|---|---|---|---|---|---|---|
| Raw | 70B-zero-shot | 80.3/56.3/66.6 | 83.6/60.6/70.2 | 83.6/60.6/69.3 | 84.0/58.0/69.1 | 82.6/57.6/68.3 | 82.6/57.6/68.3 |
| | 70B-post-trained | 85.0/62.0/71.8 | 84.6/66.3/74.1 | 84.6/66.3/75.3 | 83.0/66.0/73.2 | 81.0/65.0/71.8 | 81.0/65.0/71.8 |
| Ours | **4B-wide ⤳ 70B** | 83.6/74.6/78.5 | 84.0/75.3/79.0 | 83.0/75.3/79.1 | 82.6/75.3/78.4 | 84.0/74.6/78.6 | 84.6/73.0/78.0 |
| | **4B-depth ⤳ 70B** | 84.6/73.6/78.3 | 85.0/71.3/77.1 | 85.3/71.3/77.9 | 81.6/71.0/75.5 | 83.3/71.3/76.4 | 83.3/64.0/72.2 |
| | **8B ⤳ 70B** | 85.6/74.6/79.3 | 82.6/76.0/78.8 | 83.6/76.0/77.7 | 84.0/75.0/78.8 | 83.3/74.0/78.0 | 83.6/75.0/78.7 |
| | **8B ⤳ 405B** | 87.0/77.0/81.3 | 85.8/76.0/80.2 | 86.0/76.3/80.4 | 87.2/75.5/80.5 | 85.3/76.0/79.9 | 85.6/75.2/79.7 |

capture these post-trained behaviours and guide the pre-trained capabilities of larger LMs without compromising performance. We use 8B LMs as baselines in normal size, and we also downsize them to two 4B variants: Llama-3.1-Minitron-4B-Width, which reduces the hidden size of each layer; and Llama-3.1-Minitron-4B-Depth, which cuts the number of layer (Sreenivas et al., 2024). Despite their smaller size, they maintained comparable accuracy in weak-to-strong control. While the 4B-Width LM underperformed the 4B-Depth LM in zero-shot scenarios, its post-training results surpass the 4B-Depth, especially when controlling the 70B large LM, demonstrating its superior transferability. These results highlight two key points: *(i) downsizing the weak component can* **still effectively guide larger LMs without a significant loss in accuracy, and *(ii)* reducing model width, rather than depth, tends to be more generalizable, as deeper models demonstrate better transferability**—aligning with learning principles of the width-depth trade-offs in small-scale studies (Telgarsky, 2016; Lu et al., 2017; Raghu et al., 2017).

## 3.5 TRANSFERABILITY OF SCALED BAYESIAN INFERENCE

Although the small LMs are post-trained on the *apartment*, our overall framework is expected to be stable and generalizable across various unseen scenarios since it has the structured Bayesian framework and redirected larger LMs. To evaluate the transferability, Tab.4 compares our method with baseline models in five previously unseen scenarios: *Andersen fairy tales, ancient Egyptian, outer space, wild west, and medieval castle*. These diverse settings assess the generalisability of our approach beyond the post-training scenario. When scaling the strong component (i.e., the large controlled LMs) from 70B to 405B across these new scenarios, there are continuous improvements in ToM understanding. **This demonstrates that the increased capacity of our scaled solution enhances the transferability of ToM reasoning across multiple dynamic and unseen environments.** Furthermore, when the weak controller component is reduced from 8B to 4B, performance remains stable, ranging between 78.0% and 79.15%. This result is comparable to the 79.05% accuracy achieved in the original apartment scenario and also remains close to the performance of the 8B LMs. This consistency suggests that downsizing the weak component does not significantly affect performance, even in new and diverse test environments. **These results indicate that our approach has strong potential for continually downsizing smaller LMs as controllers since they also are capable of capturing the post-trained behaviours.** It allows saved resources to be potentially allocated to stronger controlled LMs, while still keeping stable to scenarios unseen previously.

### 3.6 WEAK-TO-STRONG CONTROL REDIRECTS LARGE LM'S LATENT REASONING

To quantify the influence of the weak controller during Bayesian inference, we analyze the changes in likelihood estimates before and after applying weak-to-strong control at each Bayesian step. As shown in Fig.2, we sample ten test cases from five datasets and average the results. It illustrates the progressively increasing magnitude of likelihood changes as Bayesian inference progresses: *(i)* At the beginning of Bayesian inference, when the large LM is close to a general initial state, the likelihood changes are minimal. This is because the general state aligns closely with pretrained world knowledge, requiring little correction from ToM-specific behaviours; *(ii)* As the model approaches a more specialized final hypothesis, the likelihood estimates are increasingly redirected. This occurs because the specialized scenarios demand ToM-specific behaviours, which the post-trained small LMs are fine-tuned to capture. The post-trained small LMs are specifically fine-tuned to the ToM context, enabling them to model human actions, goals, beliefs, and environmental states across various ToM scenarios. **Overall, this analysis finds that the weak component progressively redirects the latent reasoning of larger models, guiding them toward more accurate ToM predictions among diverse scenarios throughout the Bayesian inference process.**

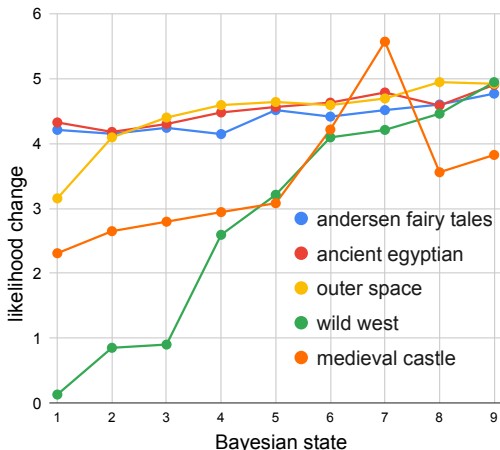

Figure 2: Likelihood change during Bayesian inference under weak-to-strong control. Results are averaged over ten sampled cases across five different unseen scenarios.

### 3.7 POST-TRAINING ALIGNS LARGE LM'S LIKELIHOOD ESTIMATION AT THE CONCEPT LEVEL

Previous experiments demonstrated that post-training on small LMs can progressively guide the behaviour of large LMs throughout Bayesian inference. Now, we further focus on how post-trained small LMs influence large LMs' likelihood estimation **at the concept level**. Fig.3 shows the execution of ten inference trials with a temperature of 0.7. The scenario involves the agent James interacting with objects in an apartment, aiming to retrieve a bottle of wine. The initial state $s_i$ is *pear in the basket, no wine*, the belief $b_i$ is *wine in the cabinet*, the goal $g_i$ is *obtain a bottle of wine*, and the action $a_i$ is *open basket, walk to cabinet*. The baseline small LM assigns lower likelihoods to fine-grained item-level concepts (e.g., wine, wine glass). After post-training,

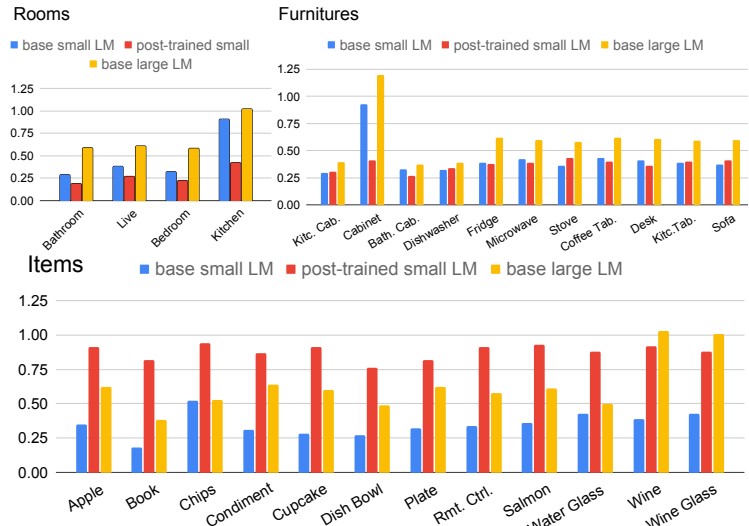

Figure 3: Likelihood estimation across different levels of concept granularity (rooms, furniture, and items) for base small LM, post-trained small LM, and base large LM. The Bayesian framework uses an LM as the **policy model** to infer actions conditioned on states, beliefs, and goals, where actions often refer to fine-grained item-level concepts (e.g., wine, wine glass). It highlights the trend of how each model allocates likelihood across these concept levels. The ToM scenario of this case is detailed at §D.7.

the small LM significantly shifts its focus toward item-level concepts, aligning its predictions more closely with the action space. This adjustment increases the likelihood assigned to critical items like wine and wine glass, which are necessary for accurately predicting the agent's goal. Consequently, post-training enables the small LM to better capture fine-grained details of the agent's behaviour, improving ToM predictions. In contrast, the large LM distributes its likelihood more evenly across all levels, from rooms to items, reflecting a broad understanding of the environment. While this approach captures general spatial awareness—identifying key areas like the kitchen and furniture like the cabinet—it lacks the sharp focus on fine-grained details, such as wine and wine glass, which are crucial for this task. As a result, the large LM may struggle with tasks that require precise, item-level predictions. Overall, post-training helps the small LM focus on item-level concepts, making it more effective for this task. While the large LM captures a broader understanding of the environment, it benefits from post-trained behaviours that redirect its likelihood estimation toward fine-grained, item-level predictions. **This finding reflects the role of post-trained small LMs in guiding large LMs' concepts toward more precise ToM reasoning.**

## 4 RELATED WORK

**Modelling human mental states** There are many studies on understanding human behaviour by classifying and predicting physical motion patterns (Aggarwal & Ryoo, 2011; Caba Heilbron et al., 2015; Choi & Savarese, 2013; Shu et al., 2015). Beyond physical behaviour, some studies focus specifically on modeling human mental states, i.e. ToM. ToM models have followed two broad approaches: Bayesian methods and end-to-end deep learning. Bayesian ToM models (Baker et al., 2017; Jara-Ettinger, 2019; Shu et al., 2021) rely on structured probabilistic frameworks to infer mental states from sparse observations of human behaviour. On the other hand, end-to-end models such as ToMnet (Rabinowitz et al., 2018; Shu et al., 2021; Sclar et al., 2022) have been trained directly on ToM tasks, learning relationships between data patterns without explicit causal models of mental states (Sap et al., 2022; Zhi-Xuan et al., 2022; Ullman, 2023). More recently, neurosymbolic reasoning systems use the neural models for feature extraction, while also incorporating probabilistic models for structured reasoning (Wong et al., 2023; Ying et al., 2024; 2023). They face challenges in dynamic and multimodal environments, where both physical and mental state reasoning are required. Different from prior studies, our work operates in more complex and dynamic multimodal ToM environments, where physical actions and mental state reasoning are intertwined.

**Post-training LLMs for downstream tasks** Post-training can project the pre-trained capabilities of LMs into downstream tasks such as dialogue generation (Ouyang et al., 2022), human value alignment (Bai et al., 2022), and multimodal tasks (OpenAI, 2023b; Liu et al., 2023). Previous approaches have also used reweighting techniques to adjust the output predictions of fine-tuned LLMs, interpolating the effects of fine-tuning with pre-trained knowledge to achieve human-centered and trustworthy text generation (Liu et al., 2021; Mitchell et al., 2024; Liu et al., 2024). More recently, LLMs are post-trained as large action/policy models for decision-making in embodied agents, allowing them to interact with and explore environments (Kim et al., 2024; Szot et al., 2024; Li et al., 2024). Our study differs by framing LMs as **policy models** in the context of Bayesian inverse inference, specifically to model human mental states. We address the limitations of existing ToM methods by scaling large policy models at test time using a likelihood redirection strategy, reasoning more accurately in complex ToM scenarios. See App.A.1 for additional discussions.

## 5 DISCUSSION AND CONCLUSION

This study investigates scalable Bayesian inference in complex and dynamic ToM environments. Existing methods based on normal-sized LMs often fail to provide sufficient reasoning capabilities and world knowledge, particularly when used as likelihood estimators in diverse challenging ToM scenarios. To overcome these limitations, we propose a method that abstracts and transfers the post-trained behavioral patterns of smaller LMs. This approach allows the extensive world knowledge of large LMs to be progressively redirected towards ToM reasoning tasks. The weak-to-strong control mechanism enables scalable reasoning by using small, post-trained LMs to guide large LMs at test time. This approach avoids additional post-training resources for large models, yet allows effective test-time scaling of Bayesian ToM reasoning even in dynamic and complex scenarios.

## ETHICS STATEMENT

This study introduces new approaches for scaling Bayesian ToM inference and is accompanied by the release of new datasets representing diverse cultural contexts. We emphasize that these datasets are either based on ancient or fictional cultures, which reduces the sensitivity to contemporary real-world issues. The content of these datasets has been carefully reviewed to avoid concerns related to discrimination, bias, and fairness. They do not involve any real individuals, ensuring that no privacy or security issues are implicated. We remain committed to responsible use and encourage continued scrutiny of potential biases or unintended consequences that may arise in future work.

## REPRODUCIBILITY STATEMENT

We are committed to ensuring that the methods in this paper are fully reproducible. The detailed descriptions of the methods, datasets, and experimental settings are in the main section and appendix, with further elaborations in the anonymous repository: https://anonymous.4open.science/r/scale-bayesian-tom-248B

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

## A    COMPARISON OF METHODOLOGIES FOR ToM INFERENCE

Table 5: Attributes of each method for ToM task.

| method | scalability | structured reasoning | world knowledge | multimodality |
|---|---|---|---|---|
| Bayesian ToM models | ✗ | ✓ | ✗ | ✗ |
| end-to-end ToM models | ✗ | ✗ | ✓ | ✓ |
| **ours** | ✓ | ✓ | ✓ | ✓ |

Tab.5 provides a comparative analysis of various methodologies for ToM inference, supplementing the discussion in the introduction (§1). Our proposed approach differs significantly from the underlying philosophies of Bayesian ToM models and end-to-end models. While Bayesian models emphasize structured reasoning guided by principles from cognitive science, they often lack scalability and struggle to handle multimodal inputs. In contrast, end-to-end models incorporate extensive world knowledge but lack the structured reasoning capabilities essential for accurate ToM inference.

Our method integrates these attributes: scalability (e.g., up to 405B), structured reasoning, robust world knowledge, and the ability to process multimodal inputs. Furthermore, our method demonstrates superior scalability, leveraging the stronger reasoning capabilities of large LMs at test time without the need for extensive post-training on large models. This allows our approach to efficiently handle complex and dynamic ToM scenarios.

### A.1    OUR THEORETICAL RATIONALES IN SCALED ToM INFERENCE AND RELATED WORK

Our approach is based on a high-level principle derived from Theorem 1 and its proof, which implies that smaller models can approximate the scaled gradient of the loss function for larger models. This mechanism bypasses direct parameter updates in the larger model, capturing the primary adjustments needed for fine-tuning while exploiting the innate generalisation capacity of the larger model. By relying on the approximate knowledge provided by the smaller model, our framework reduces computational overhead and improves scalability.

This principle is related with previous studies that have explored reweighting mechanisms for various applications (where not necessarily the same as our perspective of scaling or embodied policy model), including avoiding toxicity in text generation (Liu et al., 2021), mitigating harmful outputs in aligned models (Zhou et al., 2024), adjusting code generation (Mitchell et al., 2024), controlling sentiment in text (Han et al., 2024), and reducing hallucination or degeneration in neural text (Chuang et al., 2024; Su et al., 2022). These works demonstrate how reweighting can approximate the behaviour of large language models, mimicking direct fine-tuning in specific contexts. In contrast, our scaled ToM inference extends this principle beyond text generation tasks into the domain of social cognitive reasoning. Our framework uses language models to approximate policy behaviours for probability estimation in embodied simulators, based on the cognitive science-inspired Bayesian ToM framework. Unlike previous work focusing on text-based tasks such as sentiment or factuality control, our method addresses the unique challenges of ToM tasks, which require complex reasoning and the integration of world knowledge. These tasks involve multimodal scenarios that require understanding of agents' beliefs, goals and actions - a domain distinct from the text generation problems addressed in previous studies.

## B    DATA FLOW AND PROCESSING IN SCALABLE BAYESIAN ToM INFERENCE

### B.1    OVERALL DATA FLOW

For a detailed depiction of the data flow in our method, refer to Fig.4. The symbolic representation tools first convert video and textual descriptions into structured symbolic inputs, which are then processed by the Bayesian inference framework. This framework leverages a large LM as a scaled policy model, dynamically controlled by task-specific priors provided by a post-trained small LM, enabling accurate estimation of action likelihoods in dynamic scenarios.

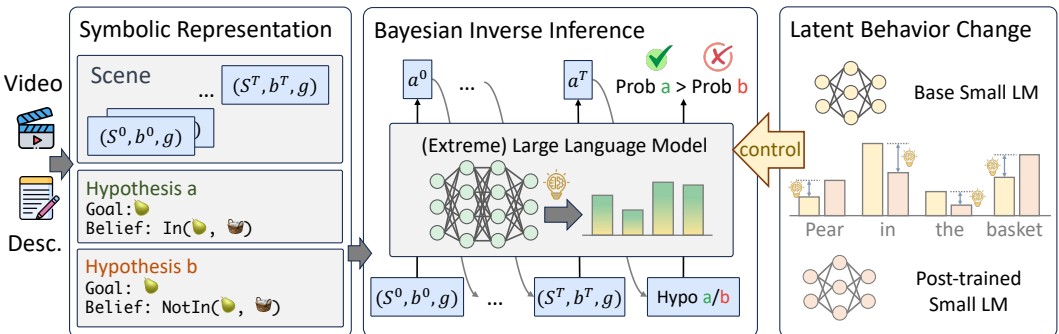

Figure 4: The data flow in our scalable Bayesian ToM inference framework. Video scenes and their corresponding descriptions are first processed by multimodal symbolic representation tools Jin et al. (2024), generating structured symbolic inputs (states, beliefs, goals). These symbolic representations are then integrated into the Bayesian inference process, where a large language model (LM) operates as a scaled **policy model** to estimate the likelihood of an agent's actions in dynamic environments. The right panel demonstrates the latent behavioral changes introduced by the post-trained small LM, which provides task-specific priors to guide the larger LM via a control mechanism.

## B.2 DATA PREPROCESSING: UNIFIED SYMBOLIC REPRESENTATIONS

To enable Bayesian ToM inference at scale, following established methods mentioned in MMToM (Jin et al., 2024; Blukis et al., 2022), multimodal data (video and textual descriptions) are transformed into structured symbolic representations. This process involves three key components: **visual perception**, **text parsing**, and **information fusion**. Together, these components provide a unified representation of states, actions, and hypotheses required for ToM tasks.

**Visual Perception.** The visual perception module is designed to process video frames and extract symbolic representations of the environment. For each frame, a scene graph is generated to capture the spatial and relational properties of objects and agents with the scene graph generator(Blukis et al., 2022). Following established methods in MMToM (Jin et al., 2024), voxel maps and 3D bounding boxes are utilized to infer object positions, containment relationships, and human poses. For instance, objects such as *pear* and *basket* are represented by predicates like In(pear, basket). These predicates effectively summarize the physical state of the environment, serving as critical inputs for subsequent reasoning steps.

**Text Parsing.** To extract symbolic representations from textual descriptions, one LLM (e.g., GPT-4) processes the text into three distinct components: (i) the initial state of the environment, (ii) human actions, and (iii) the question. Each component is translated into symbolic predicates. For example:

- The **state** is represented as predicates like In(pear, basket).

- The **action** is represented as commands such as walk towards kitchen.

- The **question** is decomposed into two hypotheses, each comprising a goal (e.g., pear) and a belief (e.g., In(pear, basket) or its negation, ¬In(pear, basket)).

This symbolic parsing ensures compatibility with the structured reasoning framework.

**Fusion.** The fusion module integrates symbolic information from video and text into a unified representation. First, predicates extracted from video inputs (e.g., spatial relationships) are aligned with those parsed from text to form the **initial state**. Next, human actions detected from the video are matched with text-based actions, and the video sequence is segmented into discrete time steps corresponding to these actions. Starting from the initial state, the symbolic representation of the environment is updated at each time step based on newly detected predicates. This process results in a sequence of symbolic states and actions, which serve as the input for Bayesian inference. Additionally, the parsed question provides two hypotheses—goal and belief—that guide the reasoning task.

## C  THEORETICAL RATIONALE ON SCALING BAYESIAN TOM INFERENCE

**Theorem 1.** *Let $\pi^{\mathcal{L}}$ be a pretrained base model, $\pi^{\mathcal{E}}$ and $\pi^{\mathcal{N}}$ be smaller tunable models where $\pi^{\mathcal{E}}$ is fine-tuned on the target task, and $\pi^*$ be the directly tuned base model. Suppose the logit adjustment $\Delta s(X_t|x_{<t}) = s_{\pi^{\mathcal{E}}}(X_t|x_{<t}) - s_{\pi^{\mathcal{N}}}(X_t|x_{<t})$ approximates the scaled negative gradient of the cross-entropy loss for logits, i.e.,*

$$\Delta s \approx -\eta \nabla_s \mathcal{L}_{CE}(s_{\pi^{\mathcal{L}}}, y), \tag{7}$$

*where $\eta$ is the learning rate. Then, the proxy-tuned model $\tilde{\pi}$, defined by*

$$s_{\tilde{\pi}}(X_t|x_{<t}) = s_{\pi^{\mathcal{L}}}(X_t|x_{<t}) + \Delta s(X_t|x_{<t}), \tag{8}$$

*approximates the directly tuned base model $\pi^*$. The KL divergence between their output distributions has this relation:*

$$D_{KL}(P_{\pi^*} \| P_{\tilde{\pi}}) \leq \frac{\eta^2}{2} \lambda_{\max} \|\nabla_s \mathcal{L}_{CE}(s_{\pi^{\mathcal{L}}}, y)\|_2^2 + \mathcal{O}(\eta^3), \tag{9}$$

*where $\lambda_{\max}$ is the maximum eigenvalue of the Hessian of the cross-entropy loss for the logits.*

*Proof.* When the learning rate $\eta$ is small, and the cross-entropy loss $\mathcal{L}_{CE}$ is smooth and twice differentiable with respect to the logits $s$, then the logit adjustment $\Delta s$ approximates the scaled negative gradient of the loss as:

$$\Delta s \approx -\eta \nabla_s \mathcal{L}_{CE}(s_{\pi^{\mathcal{L}}}, y). \tag{10}$$

The logits of the directly tuned base model $\pi^*$ after fine-tuning are updated using gradient descent:

$$s_{\pi^*} = s_{\pi^{\mathcal{L}}} - \eta \nabla_s \mathcal{L}_{CE}(s_{\pi^{\mathcal{L}}}, y) + \frac{\eta^2}{2} H_s(\nabla_s \mathcal{L}_{CE}(s_{\pi^{\mathcal{L}}}, y)) + \mathcal{O}(\eta^3), \tag{11}$$

where $H_s$ is the Hessian of $\mathcal{L}_{CE}$ with respect to the logits. The logits of the proxy-tuned model $\tilde{\pi}$ are:

$$s_{\tilde{\pi}} = s_{\pi^{\mathcal{L}}} + \Delta s. \tag{12}$$

When $\Delta s \approx -\eta \nabla_s \mathcal{L}_{CE}(s_{\pi^{\mathcal{L}}}, y)$, we have:

$$s_{\tilde{\pi}} \approx s_{\pi^{\mathcal{L}}} - \eta \nabla_s \mathcal{L}_{CE}(s_{\pi^{\mathcal{L}}}, y). \tag{13}$$

The difference in logits between the directly tuned model and the proxy-tuned model is:

$$\epsilon_s = s_{\pi^*} - s_{\tilde{\pi}}. \tag{14}$$

Then we consider their expressions:

$$\epsilon_s \approx \frac{\eta^2}{2} H_s(\nabla_s \mathcal{L}_{CE}(s_{\pi^{\mathcal{L}}}, y)) + \mathcal{O}(\eta^3). \tag{15}$$

The KL divergence between the output distributions of $\pi^*$ and $\tilde{\pi}$ is constrained using the properties of the softmax function and the Lipschitz continuity of the KL divergence:

$$D_{KL}(P_{\pi^*} \| P_{\tilde{\pi}}) \leq \frac{1}{2} \|\epsilon_s\|_2^2. \tag{16}$$

Using the norm of $\epsilon_s$:

$$\|\epsilon_s\|_2^2 \approx \frac{\eta^4}{4} \|H_s(\nabla_s \mathcal{L}_{CE}(s_{\pi^{\mathcal{L}}}, y))\|_2^2. \tag{17}$$

The Hessian's norm is constrained by its maximum eigenvalue:

$$\|H_s(\nabla_s \mathcal{L}_{CE})\|_2 \leq \lambda_{\max} \|\nabla_s \mathcal{L}_{CE}(s_{\pi^{\mathcal{L}}}, y)\|_2, \tag{18}$$

which gives:

$$\|\epsilon_s\|_2^2 \leq \frac{\eta^4}{4} \lambda_{\max}^2 \|\nabla_s \mathcal{L}_{CE}(s_{\pi^{\mathcal{L}}}, y)\|_2^2. \tag{19}$$

Finally, the KL divergence is:

$$D_{KL}(P_{\pi^*} \| P_{\tilde{\pi}}) \leq \frac{\eta^2}{2} \lambda_{\max} \|\nabla_s \mathcal{L}_{CE}(s_{\pi^{\mathcal{L}}}, y)\|_2^2 + \mathcal{O}(\eta^3). \tag{20}$$

$\square$

For theoretical implications for practical applicability, this analysis demonstrates that the weak-to-strong control mechanism relies on the learned $\Delta s$ to approximate the scaled gradient $-\eta \nabla_{\_}s\mathcal{L}_{\_}\text{CE}(s_{\_}\pi^{\mathcal{L}}, y)$ with higher-order terms contributing to the residual error. Importantly, our method does not require the small LM ($\pi^{\mathcal{E}}$) to strictly approximate the exact gradient of the cross-entropy loss for the large model. Instead, the large model ($\pi^{\mathcal{L}}$) leverages its intrinsic capacity for generalization and adaptation, based only on the approximate adjustment $\Delta s$ learned by the small LM.

This inherent flexibility allows the large model to harness its pre-trained potential, activated by the weak-to-strong control mechanism, to effectively adapt to the current ToM task. Consequently, our method achieves stable advanced performance even in novel scenarios where the small LM provides only a coarse approximation of the gradient. This significantly reduces the reliance on strict fine-tuning and maximizes computational efficiency, ensuring the approach is both scalable and practical for the physical VirtualHome environment.

# D EXPERIMENTAL DETAILS

## D.1 BELIEF AND GOAL INFERENCE TYPES AND THEIR CHARACTERISTICS TO LMS

MMToM apartment scenario questions are split into seven types, assessing ToM reasoning (Jin et al., 2024): Belief Inference includes 50% of questions on True Belief (*Type 1.1*), False Belief (*Type 1.2*), and Long-Term Belief Tracking (*Type 1.3*). Goal Inference covers the remaining 50% on True Belief (*Type 2.1*), False Belief (*Type 2.2*), Updated Belief (*Type 2.3*), and Future Actions (*Type 2.4*).

Short-term Belief Inference relies heavily on world knowledge, making it more responsive to enhancements from large LMs' pretrained capabilities. In contrast, long-term reasoning—both for Belief and Goal Inference—focuses on the dynamic nature of the environment and benefits from post-training specifically aligned to ToM scenarios.

## D.2 POST-TRAINING CONFIGURATIONS

Tab.6 summarizes the LoRA post-training configurations applied to Llama-2, Llama-3, and Llama-3.1 models during policy model training. We carefully adjust $\alpha$, rank, and other hyperparameters to optimize performance across different model sizes. Notably, following prior engineering studies, a higher $\alpha$ and rank are used for smaller models (7B and 8B), while reduced values are employed for the larger 70B model to ensure efficient adaptation without overfitting.

### D.2.1 FINE-TUNING PROCESS AND RESOURCES

The fine-tuning process for smaller models (e.g., Llama-3.1-8B) was conducted using a single NVIDIA H100 GPU, leveraging BF16 mode to optimize memory usage and maintain GPU memory consumption under 60GB. This configuration enabled efficient training of policy models tailored for Theory of Mind (ToM) tasks. The fine-tuning process was executed with the following parameters:

Table 6: LoRA configuration settings for Llama-2, Llama-3, and Llama-3.1 during post-training for policy models.

| configs | 7B | 8B | 70B |
|---|---|---|---|
| bias | none | none | none |
| fan-in fan-out | false | false | false |
| inference mode | true | true | true |
| LoRA initialization | true | true | true |
| $\alpha$ | 32 | 32 | 16 |
| dropout | 0.05 | 0.05 | 0.05 |
| rank | 16 | 16 | 8 |
| target modules | [q-proj, v-proj] | | |
| task type | causal-lm | | |

- **Batch size:** 16 (achieved via a per-device batch size of 4 and gradient accumulation steps of 4),

- **Learning rate:** $5 \times 10^{-5}$,

- **Number of epochs:** 3.

Under this setup, the fine-tuning process required approximately 8 hours to converge.

### D.2.2 DATASET SIZE

The training pool size $N$ for optimizing Equation 5 was set to 20,000 data points, sourced from the MMToM dataset's training split and our released data sampled from an embodied simulator. For tasks involving transfer to new themes, the training dataset size remained consistent at 20,000 data points, ensuring a fair and uniform setup across different experiments.

### D.3 COMPARISON OF FINE-TUNING METHODS ON MMToM TASKS

To evaluate the relative performance of full fine-tuning (FFT) and LoRA fine-tuning, we conducted experiments on two smaller models, GPT2-large (Radford et al., 2019) (774M parameters) and Gemma-2B (2B parameters) Team et al. (2024). Each model was fine-tuned using datasets of 20,000 and 8,000 datapoints, over two epochs, on 8 NVIDIA A100 80GB GPUs. The results are summarised in Table 7.

Table 7: Comparison of full fine-tuning (FFT) and LoRA fine-tuning for GPT2-large and Gemma-2B across different MMToM data sizes.

|  | Fine-tuning Method | Data Size | Model Size | Accuracy (%) |
|---|---|---|---|---|
| GPT2-large | FFT | 20,000 | 774M | 63.4 |
| GPT2-large | LoRA | 20,000 | 774M | 62.4 |
| GPT2-large | FFT | 8,000 | 774M | 62.8 |
| GPT2-large | LoRA | 8,000 | 774M | 62.1 |
| Gemma-2B | FFT | 20,000 | 2B | 68.8 |
| Gemma-2B | LoRA | 20,000 | 2B | 68.5 |
| Gemma-2B | FFT | 8,000 | 2B | 67.5 |
| Gemma-2B | LoRA | 8,000 | 2B | 67.3 |

The results show several important trends. First, when sufficient training data is available (e.g. 20,000 data points), full fine-tuning consistently outperforms LoRA, with accuracy gains of 0.9-1.2 percentage points. This suggests that full training is better at exploiting richer data, especially for smaller models. Second, the performance gap between FFT and LoRA narrows for larger models. For example, Gemma-2B shows minimal differences between FFT and LoRA (0.3 percentage points on 20,000 data points), suggesting that larger models are more robust to LoRA's parameter efficiency constraints. Finally, the influence of dataset size is evident: while FFT shows greater improvements over LoRA on smaller datasets, LoRA maintains competitive performance in resource-constrained scenarios, especially for larger models.

The results show several important trends. First, when sufficient training data is available (e.g., 20,000 data points), full fine-tuning consistently outperforms LoRA, with accuracy gains of 0.9-1.2 percentage points. This suggests that full training is better at exploiting richer data, especially for smaller models. Second, the performance gap between FFT and LoRA narrows for larger models. For example, Gemma-2B shows minimal differences between FFT and LoRA (0.3 percentage points on 20,000 data points), suggesting that larger models are more robust to LoRA's parameter efficiency constraints. Finally, the influence of dataset size is evident: while FFT shows greater improvements over LoRA on smaller datasets, LoRA maintains competitive performance in resource-constrained scenarios, especially for larger models. Table 8 further demonstrates the robustness of weak-to-strong control when transferring ToM-specific fine-tuning knowledge from a smaller model (Minitron-4B-Width) to a larger model (Llama-3.1-70B). The difference in accuracy between FFT and LoRA is only 0.15 percentage points when weak-to-strong control is applied, indicating that the mechanism

Table 8: Comparison of weak-to-strong control for Llama-3.1-Minitron-4B-Width and Llama-3.1-70B using different fine-tuning methods on the smaller model.

|  | Fine-tuning Method | Data Size | Model Size | Accuracy (%) |
|---|---|---|---|---|
| Llama-3.1-Minitron-4B-Width | FFT | 20,000 | 4B | 77.00 |
| Llama-3.1-Minitron-4B-Width | LoRA | 20,000 | 4B | 76.90 |
| *Weak-to-strong control results:* |  |  |  |  |
| 4B-Width ↬ Llama-3.1-70B | FFT-trained 4B | 20,000 | 70B | 78.67 |
| 4B-Width ↬ Llama-3.1-70B | LoRA-trained 4B | 20,000 | 70B | 78.52 |

is highly effective at bridging the gap between fine-tuning methods. Importantly, this highlights the ability of the proposed method to scale ToM-specific behaviors efficiently, leveraging both computationally intensive FFT and parameter-efficient LoRA.

Overall, these experiments highlight a trade-off between computational efficiency and performance gains. Full fine-tuning achieves modest but consistent improvements, particularly for smaller models and larger datasets. However, for larger models, LoRA provides an effective alternative with near-parity in performance and significantly reduced computational overhead. Furthermore, our weak-to-strong control mechanism demonstrates stability to fine-tuning methods, enabling scalable ToM-specific behavior elicitation with high accuracy in larger models.

### D.4 IMPACT OF PRE-TRAINING QUALITY ON MMTOM TASKS

The differences in performance between the Llama-2, Llama-3 and Llama-3.1 models provide insight into the role of pre-training quality, especially at large model scales. Based on the experimental results in Table 2 and Table 3, the influence of pre-training quality diminishes primarily due to a ceiling effect, but this is only observed when comparing models within the same scale, such as the 70B parameter range. However, when comparing smaller models to larger ones, the effect of pre-training is more pronounced. For example, moving from Llama-2 7B to Llama-2 70B after ToM-specific post-training leads to a 6% improvement in belief inference accuracy (from 76.33% to 82.33%) and a 2% improvement in goal inference accuracy (from 70% to 72%), highlighting the role of scaling in encoding richer representations.

When examining why pre-training becomes less effective at larger scales, such as comparing Llama-2-70B (pre-trained with 2.2 trillion tokens) to Llama-3.1-70B (pre-trained with 15 trillion tokens), the results suggest that larger pre-training corpora improve performance primarily for tasks that rely heavily on world knowledge: Tasks involving belief inference, which rely on short-term reasoning and general world knowledge, show significant improvements due to improved representations learned during pre-training. For example, Llama-3.1 achieves a 3.67% improvement in belief inference accuracy over Llama-2 (from 83.00% to 85.67%). These tasks benefit from richer pre-training datasets that refine the model's understanding of common human behaviours and object interactions.

In contrast, goal inference tasks that rely on long-term reasoning, including integrating temporal observations and dynamically updating beliefs, show smaller gains from larger pre-training corpora. For example, Llama-3.1 improves goal inference accuracy by only 1.67% over Llama-2 (from 72.33% to 74.00%). Such tasks are more dependent on the fine-tuning stage and the use of task-specific reasoning frameworks, such as weak-to-strong control. These results suggest that for complex reasoning tasks, the primary performance bottleneck shifts from pre-training quality to the reasoning strategies employed during fine-tuning.

In summary, pre-training quality has a significant impact on smaller models and tasks that rely heavily on world knowledge, such as belief inference. However, as models scale up to 70B parameters, the influence of pre-training diminishes due to ceiling effects, and logical reasoning tasks such as goal inference rely more on task-specific adaptations during fine-tuning.

### D.5 HOW CONSISTENT IS THEORY OF MIND ACROSS DIFFERENT PHRASINGS?

As shown in Table 4, the "*All*" column across different themes (e.g. Apartment, Andersen Fairy Tales, etc.), there is noticeable performance variance even within models of the same scale. To quantify this,

we measured the range of variance for three configurations: **70B-zero-shot**, **70B-post-trained** and **8B ↬ 70B**: (1) For 70B-zero-shot, performance ranged from 66.62 to 70.52 across themes, yielding a variance range of **3.90**; (2) For 70B-post-trained, the variance range of post-trained LMs is **3.47**, with performance ranging from 71. 86% and 75.33%; (3) For our solution 8B ↬ 70B, the weak-to-strong control mechanism further stabilised the performance, reaching only the smallest variance range of **1.62**, with scores between 77.76% and 79.38%.

These results suggest that specific topics have different effects on ToM skills, but our solution demonstrates relative stability to distributional changes caused by topic shifts. For example, **70B-zero-shot** achieves its highest performance up to 70.52% and its lowest up to 66.62%, highlighting the model's pronounced sensitivity to thematic variations in reasoning trajectories without adaptation. In contrast, our proposed solution, **8B ↬ 70B**, significantly reduces this gap, demonstrating the effectiveness of the weak-to-strong control mechanism in adjusting the ToM behaviour of the larger model while preserving the framework's general reasoning capacity across diverse and scenario-agnostic contexts.

### D.6 ON THE ROLE OF THE WEAK-TO-STRONG FRAMEWORK

The weak-to-strong framework presented in this paper focuses on aligning the larger model's distribution with ToM-specific beliefs and task structures while preserving its general reasoning capabilities, rather than primarily relying on the smaller model's reasoning abilities. This design enables efficient transfer of ToM-specific task structures without compromising the broader capabilities of the larger model.

The smaller model (e.g., 4B or 8B parameters) undergoes ToM-specific post-training to encode task-relevant priors, such as belief states and potential goals, without requiring advanced independent reasoning capabilities. During inference, the smaller model functions as an assistive scaffold, conditioning the larger model's likelihood estimation in a Bayesian framework. This role is formalized through the adjustment ratio: $\frac{\pi^{\mathcal{E}}}{\pi^{\mathcal{N}}}$, where $\pi^{\mathcal{E}}$ is the post-trained smaller model's task-specific policy, and $\pi^{\mathcal{N}}$ is the naive pre-trained smaller model's policy.

The larger model (e.g., 70B parameters) integrates this adjustment ratio to refine its likelihood estimation dynamically. The overall policy distribution is computed as $\pi^{\mathcal{L}} \frac{\pi^{\mathcal{E}}}{\pi^{\mathcal{N}}}$, where $\pi^{\mathcal{L}}$ is the policy from the larger model. This mechanism allows the larger model to retain its broad reasoning and world knowledge, ensuring its capacity for generalization while aligning with ToM-specific task structures.

To validate this framework, we compared the performance of the 8B ↬ 70B model to the 70B-post-trained model across five unseen themes, including *Andersen Fairy Tales*, *Ancient Egyptian*, and *Outer Space*. As shown in Table 9, the weak-to-strong mechanism achieved consistent improvements across all ToM tasks, demonstrating its ability to preserve and transfer the larger model's general reasoning capabilities while aligning with ToM-specific requirements. These results, combined with theoretical

Table 9: Performance of the 8B ↬ 70B model on unseen themes compared to 70B-post-trained and 70B-zero-shot models across all ToM tasks.

| Unseen Theme | Scale | 1.1 | 1.2 | 1.3 | Avg. | 2.1 | 2.2 | 2.3 | 2.4 | Avg. | All |
|---|---|---|---|---|---|---|---|---|---|---|---|
| **Andersen Fairy Tales** | 70B-zero-shot | 88.00 | 73.00 | 90.00 | 83.67 | 70.67 | 80.00 | 25.33 | 66.67 | 60.67 | 70.52 |
| | 70B-post-train | 90.00 | 71.00 | 93.00 | 84.67 | 73.33 | 61.33 | 61.33 | 69.33 | 66.33 | 74.19 |
| | 8B ↬ 70B | 92.00 | 71.00 | 85.00 | 82.67 | 82.67 | 76.00 | 68.00 | 77.33 | 76.00 | 78.86 |
| **Ancient Egyptian** | 70B-zero-shot | 89.00 | 71.00 | 91.00 | 83.67 | 74.67 | 74.67 | 25.33 | 60.00 | 58.67 | 69.38 |
| | 70B-post-train | 89.00 | 69.00 | 96.00 | 84.67 | 72.00 | 76.00 | 61.33 | 64.00 | 68.33 | 75.33 |
| | 8B ↬ 70B | 90.00 | 73.00 | 88.00 | 83.67 | 69.33 | 76.00 | 73.33 | 74.67 | 73.33 | 77.76 |
| **Outer Space** | 70B-zero-shot | 88.00 | 72.00 | 92.00 | 84.00 | 72.00 | 64.00 | 25.33 | 70.67 | 58.00 | 69.38 |
| | 70B-post-train | 91.00 | 68.00 | 90.00 | 83.00 | 69.33 | 65.33 | 61.33 | 68.00 | 66.00 | 75.33 |
| | 8B ↬ 70B | 90.00 | 70.00 | 92.00 | 84.00 | 73.33 | 81.33 | 66.67 | 78.67 | 75.00 | 77.76 |

insights from Section C, demonstrate that the weak-to-strong framework effectively utilizes the smaller model as a task-specific lens to guide the larger model's predictions. This collaborative dynamic ensures alignment with ToM-specific task requirements while preserving general reasoning capabilities.

Table 10: Detailed transfer performance of the Bayesian method with different scaling strategies (zero-shot, direct post-training, and our weak-to-strong control) from the original apartment scenario to various unseen environments. All models are based on Llama3.1.

| Theme | Scale | Belief Inference | | | | Goal Inference | | | | | All |
|---|---|---|---|---|---|---|---|---|---|---|---|
| | | 1.1 | 1.2 | 1.3 | Avg. | 2.1 | 2.2 | 2.3 | 2.4 | Avg. | |
| Andersen fairy tales | 70B-zeroshot | 88.00 | 73.00 | 90.00 | 83.67 | 70.67 | 80.00 | 25.33 | 66.67 | 60.67 | 70.52 |
| | 70B-post-train | 90.00 | 71.00 | **93.00** | 84.67 | 73.33 | 61.33 | 61.33 | 69.33 | 66.33 | 74.19 |
| | 8B ↝ 70B | **92.00** | 71.00 | 85.00 | 82.67 | **82.67** | 76.00 | 68.00 | **77.33** | **76.00** | 78.86 |
| | 4B-width ↝ 70B | 90.00 | 73.00 | 89.00 | 84.00 | 80.00 | **81.33** | **76.00** | 64.00 | 75.33 | 79.05 |
| | 4B-depth ↝ 70B | 91.00 | **74.00** | 90.00 | **85.00** | 74.67 | 73.33 | 64.00 | 73.33 | 71.33 | 77.19 |
| ancient Egyptian | 70B-zeroshot | 89.00 | **71.00** | 91.00 | 83.67 | 74.67 | 74.67 | 25.33 | 60.00 | 58.67 | 69.38 |
| | 70B-post-train | 89.00 | 69.00 | 96.00 | 84.67 | 72.00 | 76.00 | 61.33 | 64.00 | 68.33 | 75.33 |
| | 8B ↝ 70B | 90.00 | 73.00 | 88.00 | 83.67 | 69.33 | 76.00 | 73.33 | 74.67 | 73.33 | 77.76 |
| | 4B-width ↝ 70B | 90.00 | 69.00 | 90.00 | 83.00 | 70.67 | **80.00** | **85.33** | 69.33 | **76.33** | 79.19 |
| | 4B-depth ↝ 70B | 91.00 | 69.00 | **96.00** | **85.33** | **76.00** | 68.00 | 69.33 | **76.00** | 72.33 | 77.90 |
| outer space | 70B-zeroshot | 88.00 | 72.00 | 92.00 | **84.00** | 72.00 | 64.00 | 25.33 | 70.67 | 58.00 | 69.38 |
| | 70B-post-train | **91.00** | 68.00 | 90.00 | 83.00 | 69.33 | 65.33 | 61.33 | 68.00 | 66.00 | 75.33 |
| | 8B ↝ 70B | 90.00 | 70.00 | **92.00** | **84.00** | 73.33 | **81.33** | 66.67 | **78.67** | 75.00 | 77.76 |
| | 4B-width ↝ 70B | 90.00 | 70.00 | 88.00 | 82.67 | **73.33** | 76.00 | **80.00** | 72.00 | **75.33** | 79.19 |
| | 4B-depth ↝ 70B | 90.00 | 69.00 | 86.00 | 81.67 | 70.67 | 73.33 | 68.00 | 72.00 | 71.00 | 77.90 |
| wild west | 70B-zeroshot | 88.00 | 72.00 | **92.00** | **84.00** | 72.00 | 64.00 | 25.33 | 70.67 | 58.00 | 69.14 |
| | 70B-post-train | **91.00** | 68.00 | 90.00 | 83.00 | 69.33 | 65.33 | 61.33 | 68.00 | 66.00 | 73.29 |
| | 8B ↝ 70B | 90.00 | 70.00 | **92.00** | **84.00** | 73.33 | **81.33** | 66.67 | **78.67** | 75.00 | **78.86** |
| | 4B-width ↝ 70B | 90.00 | 70.00 | 88.00 | 82.67 | **73.33** | 76.00 | **80.00** | 72.00 | **75.33** | 78.48 |
| | 4B-depth ↝ 70B | 90.00 | 69.00 | 86.00 | 81.67 | 70.67 | 73.33 | 68.00 | 72.00 | 71.00 | 75.57 |
| medieval castle | 70B-zeroshot | 88.00 | **71.00** | 89.00 | 82.67 | 62.67 | 74.67 | 20.00 | 73.33 | 57.67 | 68.38 |
| | 70B-post-train | 85.00 | 69.00 | 89.00 | 81.00 | 65.33 | 69.33 | 57.33 | 68.00 | 65.00 | 71.86 |
| | 8B ↝ 70B | 90.00 | 72.00 | 89.00 | 83.67 | 72.00 | 76.00 | 68.00 | **84.00** | 75.00 | **78.71** |
| | 4B-width ↝ 70B | **92.00** | 71.00 | **91.00** | **84.67** | **77.33** | **77.33** | **69.33** | 68.00 | **73.00** | 78.00 |
| | 4B-depth ↝ 70B | 90.00 | 70.00 | 90.00 | 83.33 | 58.67 | 72.00 | 53.33 | 72.00 | 64.00 | 72.29 |

## D.7 THEORY-OF-MIND CASE STUDY: AGENT JAMES IN APARTMENT INTERACTION

Fig.5 provides a detailed visual and language-based description of the test case described in experiment §3.7 of the experiment, where the likelihood estimation behaviour of different LMs is discussed across varying concept levels.

## D.8 TOM TRANSFER EFFECT ON UNSEEN SCENARIOS

Tab.10 supplements the results in experiment §3.5, providing a detailed comparison between the baselines and our scalable solution across belief inference and goal inference subtasks in various unseen ToM scenarios. Our experimental observations are consistent with those outlined in §3.5: *(i)* The increased capacity of our scalable solution significantly improves the transferability of ToM reasoning across dynamic and previously unseen environments. *(ii)* Our approach demonstrates strong potential for downsizing small LMs as controllers, as they successfully capture the post-trained behaviours and exhibit robust performance in guiding larger models. *(iii)* Notably, our method can approximate—and in some cases outperform—the results achieved by directly post-training large-scale LMs (such as the 70B model). These findings underscore the flexibility and scalability of our approach for handling practical ToM tasks in diverse, complex environments.

## D.9 THEMATIC SCENARIO DATA FOR TOM TASK TRANSFER

As described in §D.8, five new thematic scenarios are used for evaluation: Andersen Fairy Tales, Ancient Egyptian, Wild West, Outer Space and Medieval Castle. These environments are not seen during the post-training phase of our method and are different from the original *apartment* setting.

The transfer to these scenarios demonstrates the generalisability of our solution to dynamically adapt to different domains, with each thematic environment presenting unique challenges and contextual shifts from the apartment scenario. Fig.6 provides a visual summary of these key differences, statistically extracted and mapped to illustrate the transformation of concept and environment across these themes. These distinctions are used to evaluate ToM task transfer across different dynamic environments.

Figure 5: Theory-of-Mind scenario used in the main experiments §3.7, involving an agent (James) interacting with objects in an apartment.

Figure 6: Primary changes between the original apartment scenario and the five transferred thematic environments used in our ToM experiments.

