# OpenReview forum: "Scaling Multimodal Theory-of-Mind with Weak-to-Strong Bayesian Reasoning"
_ICLR.cc/2025/Conference — Submitted to ICLR 2025_

### Official Review · Reviewer_CTGe · 2024-10-31

**Soundness:** 3
**Presentation:** 2
**Contribution:** 3
**Rating:** 6
**Confidence:** 1

**Summary:**

The paper introduces a novel approach to utilizing large-scale models for Bayesian Theory of Mind (ToM) reasoning. Rather than fine-tuning large models directly, the authors propose fine-tuning smaller models and leveraging the difference between the fine-tuned and naive versions of these small models to approximate the impact of fine-tuning on the larger model. This method is rigorously evaluated on the MMToM-QA dataset, demonstrating its effectiveness in handling complex ToM tasks.

In the interest of transparency, I would like to note that this paper is somewhat outside my area of expertise, and as such, my confidence in assessing it is limited. However, after discussing this with the Area Chair, we agreed that I would proceed with the review and focus on evaluating aspects I can assess.

**Strengths:**

The proposed method is both simple and robust, demonstrating a well-grounded approach. The experiments are rigorously conducted, and the results reflect strong performance, effectively supporting the validity of the approach.

**Weaknesses:**

A weakness I noted is that I found the paper challenging to understand. While this is possibly due to my limited expertise in this specific area, I looked at recent related works (BIP-ALM and SimToM) and found them notably more accessible. This suggests that, although my background may play a role, the paper could benefit from clearer explanations or added context to help readers better grasp its motivation and contribution - especially those outside the immediate field.

**Questions:**

- The general idea of approximating the changes that fine-tuning a large model would have by fine-tuning a smaller one and approximating the changes by reweighing using a ratio seems very general. Could this exact methodology be applied to other types of problems? (Or was it already used? - the relation to the works using reweighing, mentioned in the related work, is unclear to me. )

- Is there a fundamental reason why fine-tuning models for the kind of ToM tasks considered is harder / more computationally expensive than fine-tuning for other tasks?

---

> ### Author Response · Authors · 2024-11-22
> **Response to Reviewer CTGe (1/5)**
>
> We thank Reviewer CTGe for their insightful review and comments. We have addressed their feedback below, and these updates are also reflected in the revised version of our paper.
>
> ---
>
> **[W1: A weakness I noted is that I found the paper challenging to understand. While this is possibly due to my limited expertise in this specific area, I looked at recent related works (BIP-ALM and SimToM) and found them notably more accessible. This suggests that, although my background may play a role, the paper could benefit from clearer explanations or added context to help readers better grasp its motivation and contribution—especially those outside the immediate field.]**
>
> Thanks for the feedback regarding the clarity of our paper. We have revised the manuscript, drawing inspiration from BIP-ALM and SimToM, to improve accessibility and ensure broader understanding of our motivation and contributions.
>
> Regarding the context of Theory-of-Mind, we have revised and simplified the introduction (lines 24–31) to provide a more accessible explanation of Theory-of-Mind (ToM) and its significance in AI research. The revised section is as follows:
>
> > "A key aspect of human social cognition is Theory-of-Mind (ToM)—the ability to comprehend and attribute mental states such as beliefs, desires, and intentions to ourselves and others. This capacity allows individuals to recognize that others may have perspectives and motivations distinct from their own, forming the foundation of social understanding and interaction (Dennett, 1988; Gopnik & Wellman, 2012). Building on this concept, a critical challenge lies in enabling artificial intelligence (AI) to acquire human-level ToM capabilities. Equipping AI systems with such abilities could significantly enhance their potential for human-like interactions and unlock broader capacities for commonsense and context-aware reasoning (Lake et al., 2017; Wu et al., 2021; Ma et al., 2023)."
>
> Regarding the motivation, we have explicitly added a detailed explanation of the motivation behind our work, emphasizing the need for scalable Bayesian ToM inference in large language models. This has been included in the introduction, as follows:
>
> > "ToM intricately intertwines **open-domain world knowledge** and **implicit reasoning** to ground human mental states within their corresponding physical environments. As environments and their associated queries evolve, reliance on smaller post-trained LMs introduces a critical trade-off: while smaller LMs excel in adapting to specific ToM tasks within particular environments, larger LMs are indispensable for harnessing broader world knowledge and advanced reasoning capabilities. **This inherent tension poses a significant challenge, requiring us to improve the scalability of Bayesian ToM methods in handling dynamically evolving, multimodal ToM scenarios.**"
>
> Regarding the contribution, we have also explicitly provided a detailed explanation of our contribution within the context of Bayesian ToM reasoning, emphasizing the unique aspects of our weak-to-strong controlled scalable Bayesian ToM inference and how it differs from prior ToM solutions. This explanation has been included in the introduction, as follows:
>
> > "Motivated by the scalability challenge of current Bayesian ToM methods, we propose a scalable Bayesian inference solution that generalizes to complex and dynamic environments (Tab.5 in App.A compares our technical contributions with other solutions). Our approach introduces a weak-to-strong control mechanism where post-trained smaller LMs specialize in ToM-specific tasks by capturing likelihood inference patterns during Bayesian ToM reasoning. These ToM behaviors are then transferred to larger LMs during test time, aligning the larger models’ reasoning trajectories with the structured requirements of Bayesian inverse planning. In this framework, the larger LM acts as the primary policy model, leveraging its extensive world knowledge and reasoning capabilities. Importantly, the reasoning trajectory of the larger LM is structured by the Bayesian framework, ensuring consistency, robustness, and interpretability. This design avoids additional post-training costs for larger LMs while enabling scalable use of large models up to 70B or 405B parameters."
> >
> > **Table A: Attributes of each method for ToM tasks.**
> > | Method                 | Scalability | Structured Reasoning | World Knowledge | Multimodality |
> > |------------------------|-------------|-----------------------|-----------------|---------------|
> > | Bayesian ToM models    | ❌          | ✅                    | ❌              | ❌            |
> > | End-to-end ToM models  | ❌          | ❌                    | ✅              | ✅            |
> > | **Ours**               | ✅          | ✅                    | ✅              | ✅            |

---

> ### Author Response · Authors · 2024-11-22
> **Response to Reviewer CTGe (2/5)**
>
> > Table A provides a comparative analysis of various methodologies for ToM inference, supplementing the discussion in the introduction. Our proposed approach differs from the underlying philosophies of prior Bayesian ToM models and end-to-end models. While Bayesian models emphasize structured reasoning guided by principles from cognitive science, they often lack scalability and struggle to handle multimodal inputs. In contrast, end-to-end models incorporate extensive world knowledge but lack the structured reasoning capabilities essential for accurate ToM inference. Our method integrates these attributes: scalability (e.g., up to 405B), structured reasoning, robust world knowledge, and the ability to process multimodal inputs. Furthermore, our method demonstrates superior scalability, leveraging the stronger reasoning capabilities of large LMs at test time without the need for extensive post-training on large models. This allows our approach to efficiently handle complex and dynamic ToM scenarios."
>
> Regarding the theoretical **motivation** behind our **contribution on the technical level**, we have also provided a Theorem and Proof to explain the principle of our method in Appendix C, quoted here:
>
> > **_Theorem:_**
> > Let $\pi^\mathcal{L}$ be a pretrained base model, $\pi^\mathcal{E}$ and $\pi^\mathcal{N}$ be smaller tunable models where $\pi^\mathcal{E}$ is fine-tuned on the target task, and $\pi^*$ be the directly tuned base model. Suppose the logit adjustment $\Delta s(X_t | x_{<t}) = s_{\pi^\mathcal{E}}(X_t | x_{<t}) - s_{\pi^\mathcal{N}}(X_t | x_{<t})$ approximates the scaled negative gradient of the cross-entropy loss for logits, i.e.,
> > $$
> > \Delta s \approx -\eta \nabla\_s \mathcal{L}\_{\text{CE}}(s\_{\pi^\mathcal{L}}, y),
> > $$
> > where $\eta$ is the learning rate. Then, the proxy-tuned model $\tilde{\pi}$, defined by
> > $$
> > s\_{\tilde{\pi}}(X\_t | x\_{<t}) = s\_{\pi^\mathcal{L}}(X\_t | x\_{<t}) + \Delta s(X\_t | x\_{<t}),
> > $$
> > approximates the directly tuned base model $\pi^*$. The KL divergence between their output distributions is constrained by:
> > $$
> > D\_{\text{KL}}(P\_{\pi^*} \| P\_{\tilde{\pi}}) \leq \frac{\eta^2}{2} \lambda\_{\max} \|\nabla\_s \mathcal{L}\_{\text{CE}}(s\_{\pi^\mathcal{L}}, y)\|\_2^2 + \mathcal{O}(\eta^3),
> > $$
> > where $\lambda\_{\max}$ is the maximum eigenvalue of the Hessian of the CE loss for the logits.
> >
> > **_Proof:_**
> > When the learning rate $\eta$ is small, and the cross-entropy loss $\mathcal{L}_{\text{CE}}$ is smooth and twice differentiable with respect to the logits $s$, then the logit adjustment $\Delta s$ approximates the scaled negative gradient of the loss as:
> > $$
> > \Delta s \approx -\eta \nabla\_s \mathcal{L}\_{\text{CE}}(s\_{\pi^\mathcal{L}}, y).
> > $$
> >
> > The logits of the directly tuned base model $\pi^*$ after fine-tuning are updated using gradient descent:
> > $$
> > s\_{\pi^*} = s\_{\pi^\mathcal{L}} - \eta \nabla_s \mathcal{L}\_{\text{CE}}(s\_{\pi^\mathcal{L}}, y) + \frac{\eta^2}{2} H\_s (\nabla_s \mathcal{L}\_{\text{CE}}(s\_{\pi^\mathcal{L}}, y)) + \mathcal{O}(\eta^3),
> > $$
> > where $H\_s$ is the Hessian of $\mathcal{L}\_{\text{CE}}$ with respect to the logits.
> >
> > The logits of the proxy-tuned model $\tilde{\pi}$ are:
> > $$
> > s\_{\tilde{\pi}} = s\_{\pi^\mathcal{L}} + \Delta s.
> > $$
> > When $\Delta s \approx -\eta \nabla\_s \mathcal{L}\_{\text{CE}}(s\_{\pi^\mathcal{L}}, y)$, we have:
> > $$
> > s\_{\tilde{\pi}} \approx s\_{\pi^\mathcal{L}} - \eta \nabla\_s \mathcal{L}\_{\text{CE}}(s\_{\pi^\mathcal{L}}, y).
> > $$
> > The difference in logits between the directly tuned model and the proxy-tuned model is:
> > $$
> > \epsilon_s = s_{\pi^*} - s_{\tilde{\pi}}.
> > $$
> > Then we consider their expressions:
> > $$
> > \epsilon\_s \approx \frac{\eta^2}{2} H\_s (\nabla\_s \mathcal{L}\_{\text{CE}}(s\_{\pi^\mathcal{L}}, y)) + \mathcal{O}(\eta^3).
> > $$
> > The KL divergence between the output distributions of $\pi^*$ and $\tilde{\pi}$ is constrained using the properties of the softmax function and the Lipschitz continuity of the KL divergence:
> > $$
> > D\_{\text{KL}}(P\_{\pi^*} \| P\_{\tilde{\pi}}) \leq \frac{1}{2} \|\epsilon\_s\|\_2^2.
> > $$
> > Using the norm of $\epsilon\_s$:
> > $$
> > \|\epsilon\_s\|\_2^2 \approx \frac{\eta^4}{4} \|H\_s (\nabla\_s \mathcal{L}\_{\text{CE}}(s\_{\pi^\mathcal{L}}, y))\|\_2^2.
> > $$
> > The Hessian's norm is constrained by its maximum eigenvalue:
> > $$
> > \|H\_s (\nabla\_s \mathcal{L}\_{\text{CE}})\|\_2 \leq \lambda\_{\max} \|\nabla\_s \mathcal{L}\_{\text{CE}}(s\_{\pi^\mathcal{L}}, y)\|\_2,
> > \\
> > \|\epsilon\_s\|\_2^2 \leq \frac{\eta^4}{4} \lambda\_{\max}^2 \|\nabla\_s \mathcal{L}\_{\text{CE}}(s\_{\pi^\mathcal{L}}, y)\|\_2^2.
> > $$
> > The KL divergence is:
> > $$
> > D\_{\text{KL}}(P\_{\pi^*} \| P\_{\tilde{\pi}}) \leq \frac{\eta^2}{2} \lambda\_{\max} \|\nabla\_s \mathcal{L}\_{\text{CE}}(s\_{\pi^\mathcal{L}}, y)\|\_2^2 + \mathcal{O}(\eta^3).
> > $$

---

> ### Author Response · Authors · 2024-11-22
> **Response to Reviewer CTGe (3/5)**
>
> > $$
> > D\_{\text{KL}}(P\_{\pi^*} \| P\_{\tilde{\pi}}) \leq \frac{\eta^2}{2} \lambda\_{\max} \|\nabla\_s \mathcal{L}\_{\text{CE}}(s\_{\pi^\mathcal{L}}, y)\|\_2^2 + \mathcal{O}(\eta^3).
> > $$
> > The KL divergence between the directly tuned model $\pi^*$ and the proxy-tuned model $\tilde{\pi}$ is constrained by terms involving the learning rate $\eta$, the gradient norm $\|\nabla\_s \mathcal{L}\_{\text{CE}}\|$, and the Hessian's maximum eigenvalue $\lambda\_{\max}$.
> >
> > In terms of **theoretical implications for practical applicability**, this analysis demonstrates that the **weak-to-strong control mechanism** relies on the learned $\Delta s$ to approximate the scaled gradient $-\eta \nabla\_s \mathcal{L}\_{\text{CE}}(s\_{\pi^\mathcal{L}}, y)$ with higher-order terms contributing to the residual error. Importantly, our method does not require the small LM ($\pi^\mathcal{E}$) to strictly approximate the exact gradient of the cross-entropy loss for the large model. Instead, the large model ($\pi^\mathcal{L}$) leverages its **intrinsic capacity for generalization and adaptation**, based only on the approximate adjustment $\Delta s$ learned by the small LM.
> >
> This inherent flexibility allows the large model to harness its **pre-trained potential**, activated by the weak-to-strong control mechanism, to effectively adapt to the current ToM task. Consequently, our method achieves **stable advanced performance** even in novel scenarios where the small LM provides only a coarse approximation of the gradient. This significantly reduces the reliance on strict fine-tuning and maximizes computational efficiency, ensuring the approach is both scalable and practical for the physical VirtualHome environment.
>
> ---
>
> **[Q1: The general idea of approximating the changes that fine-tuning a large model would have by fine-tuning a smaller one and approximating the changes by reweighing using a ratio seems very general. Could this exact methodology be applied to other types of problems? (Or was it already used? - the relation to the works using reweighing, mentioned in the related work, is unclear to me. )]**
>
> Yes, this methodology can be applied to other types of problems. Additionally, there are already several works utilizing reweighting mechanisms. According to our *Theorem 1 and proof*, the high-level principle is that the smaller model can approximate the scaled gradient of the loss function for the larger model. This enables bypassing direct parameter updates in the large model while still capturing the primary adjustments needed for fine-tuning. Crucially, the large model’s inherent capacity for generalization allows it to adapt based on the approximate knowledge provided by the smaller model, reducing computational overhead and ensuring scalability.
>
> Guided by this principle, and stepping back to a broader perspective, prior studies have applied similar ideas to various domains, including toxicity avoidance in the text output [1], attacking aligned model exposure harmfulness in output text [2], adjusting code generation/factuality [3], controlling sentiment in output text [4], and reducing hallucinations/degeneration in output text [5, 6]. These works utilize reweighting mechanisms to approximate the behavior of language models as if directly fine-tuned, presenting a generalizable rationale across different contexts.
> However, our scaled Bayesian ToM inference differs fundamentally. Rooted in a cognitive-inspired Bayesian ToM framework, our method extends scalability by serving as a policy model for likelihood estimation in embodied simulators. Unlike prior works that focus on text generation tasks (e.g., code, sentiment, or toxicity control), our approach addresses the unique challenges of social cognition reasoning. ToM tasks demand intricate reasoning and integration of world knowledge, making them distinctly more complex and multimodal in nature. Our method uses LMs to approximate large language model behaviors in these complex scenarios, enabling scalable and effective inference based on Bayesian principles from cognitive science.
>
> **References**:
>
> 1. *DExperts: Decoding-Time Controlled Text Generation with Experts and Anti-Experts*, ACL 2021.
> 2. *Emulated Disalignment: Safety Alignment for Large Language Models May Backfire!*, ACL 2024 *(Outstanding Paper Award)*.
> 3. *An Emulator for Fine-Tuning Large Language Models using Small Language Models*, ICLR 2024.
> 4. *Word Embeddings Are Steers for Language Models*, ACL 2024 *(Outstanding Paper Award)*.
> 5. *DoLa: Decoding by Contrasting Layers Improves Factuality in Large Language Models*, ICLR 2024.
> 6. *A Contrastive Framework for Neural Text Generation*, NeurIPS 2022.
>
> ---

---

> ### Author Response · Authors · 2024-11-22
> **Response to Reviewer CTGe (4/5)**
>
> ---
>
> **[Q2: Is there a fundamental reason why fine-tuning models for the kind of ToM tasks considered is harder/more computationally expensive than fine-tuning for other tasks?]**
>
> *Table A: Comparison of direct full-finetuning training (FFT) and LoRA fine-tuning for GPT2-large and Gemma-2B across different data sizes.*
>
> | Model Name  | Fine-tuning Method | Data Size | Model Size | Accuracy (%) |
> | ----------- | ------------------ | --------- | ---------- | ------------ |
> | GPT2-large  | FFT                | 20,000    | 774M       | 63.4         |
> | GPT2-large  | LoRA               | 20,000    | 774M       | 62.4         |
> | GPT2-large  | FFT                | 8,000     | 774M       | 62.8         |
> | GPT2-large  | LoRA               | 8,000     | 774M       | 62.1         |
> | gemma-2b    | FFT                | 20,000    | 2B         | 68.8         |
> | gemma-2b    | LoRA               | 20,000    | 2B         | 68.5         |
> | gemma-2b    | FFT                | 8,000     | 2B         | 68.2         |
> | gemma-2b    | LoRA               | 8,000     | 2B         | 67.9         |
>
> We newly conducted the above experiments to evaluate how the fine-tuning process for ToM tasks improves the model's performance. In Table A, we observe the following trends:
>
> 1. As the training data size increases (e.g., from 8,000 to 20,000 data points), the growth in accuracy for both fine-tuning methods on ToM evaluations is not significant. This suggests that the fine-tuning process for multimodal ToM tasks is less effective with the currently available multimodal data sampled from physical environmental simulators.
>
> 2. Although conclusions from existing studies [1, 2] suggest that full fine-tuning (FFT) can absorb more knowledge than LoRA (as LoRA primarily re-combines and adjusts existing pretrained knowledge within existing pretrained capabilities), the superiority of FFT over LoRA becomes less significant as model size increases (e.g., Gemma-2B vs. GPT2-large-774M) in our ToM evaluations.
>
> Therefore, we infer that the fine-tuning process for multimodal ToM tasks is more challenging because the observed trends do not align with successful outcomes in other domains (e.g., code generation, mathematical QA). Specifically, more data for full fine-tuning does not necessarily outperform low-rank adaptation. This suggests that the currently available multimodal ToM data for evaluation is not as effective as datasets in other domains for improving model performance, implying that multimodal ToM is still underdeveloped and unable to directly enhance the model's understanding of multimodal ToM.
>
> When we refer to other studies that directly discuss the intrinsic source of ToM abilities in LLMs, we find that ToM capabilities are not derived primarily from fine-tuning data but from the extensive latent knowledge already present within the model. This highlights a unique challenge in ToM tasks: a mismatch between the existing ToM fine-tuning data and the goal of enabling LLMs to develop more advanced ToM abilities.
>
> For example, prior studies [3] suggest that ToM abilities stem from human preference data within the pretraining corpus and alignment data, such as instruction tuning and RLHF/DPO/etc. The relative discussion is quoted:
> > We suggest that the interlinked evolution and development of language and ToM may help explain what instruction-tuning adds: rewarding cooperative communication that takes into account interlocutor and context. We find that instruction-tuned LLMs from the GPT family outperform other models, and often also children. Base-LLMs are mostly unable to solve ToM tasks, even with specialized prompting.
> > For the Strange Stories, we saw that base-LLMs perform generally below child level. Most instructLLMs perform close to or above child level, particularly as items become more complex and child performance drops much more dramatically than LLM performance. Levels of deviation from the original test formulation seem to have made almost no impact for the SS, suggesting that the capacity to deal with non-literal language targeted by the Strange Stories test does generalize to novel contexts. We conclude that instruct-LLMs are quite capable at interpreting non-literal language, a skill that in humans involves ToM.
> > The gap between base- and instruct-LLMs is best summarized in Figure 4. Here we see that no base-LLM achieves child level: all LLMs approaching or exceeding child performance are larger instruct-LLMs. Our adapted prompts and insertion of correct answers for motivation questions did not make a difference. We suggest that another issue for baseLLMs, besides the prompt format, was prompt length.

---

> ### Author Response · Authors · 2024-11-22
> **Response to Reviewer CTGe (5/5)**
>
> > Evidence is emerging that most LLM capacities are learned during self-supervised pre-training (Gudibande et al., 2023; Ye et al., 2023), which suggests that base-LLMs are essentially ‘complete’ models. Yet instruction-tuning, even in small amounts (Zhou et al., 2023), adds adherence to the desired interaction format and teaches LLMs, as it were, to apply their knowledge appropriately. We see a parallel between instruction-tuning and the role for rewarding cooperative communication in human evolution and development. It has been argued extensively that human communication is fundamentally cooperative in that it relies on a basic ability and willingness to engage in mental coordination (e.g Verhagen, 2015; Grice, 1975). It is a key characteristic of the socio-cultural niche in which we evolved that, when growing up, we are constantly being rewarded for showing such willingness and cooperating with others to achieve successful communicative interactions (Tomasello, 2008). Reversely, if we do not, we are being punished, explicitly or implicitly via increasing social exclusion (David-Barrett and Dunbar, 2016). This brings us back to our context: instruction-tuning essentially rewards similar cooperative principles, but punishes the opposite, which may amount to an enhanced capacity for coordinating with an interaction partner’s perspective, in humans and LLMs alike. This is reflected in performance on ToM tasks, which are banking on this capacity too.
>
> In the study [4], the authors also discuss the source of ToM ability in LLMs and the unique challenges of ToM tasks, as quoted:
> > *We found that LLMs trained with Reinforcement Learning from Human Feedback (RLHF) (all models excluding Davinci-2) improved their ToM accuracy via in-context learning. .... In particular, Davinci-2 was the only model that was not finetuned with RLHF, and it was also the only model whose ToM performance was not increased by our prompt manipulations. It is possible that the RLHF component of the training enabled the models to exploit the in-context prompts in this setting.*
>
> As prior studies suggest, the ToM ability of LLMs is derived primarily from RLHF, instruction tuning, and extensive pretraining data, rather than from lightweight fine-tuning alone. This highlights a fundamental challenge in fine-tuning for ToM tasks: the mismatch between the limited fine-tuning data available and the intrinsic complexity of the knowledge and reasoning required for ToM. With this insight, **Table B** examines whether eliciting the pretrained capabilities of large LMs can significantly enhance ToM performance.
>
> **Table B: Comparison of Llama-3.1-70B controlled by direct full-finetuning training (FFT) and LoRA fine-tuning for Llama-3.1-Minitron-4B-Width via our weak-to-strong control.**
>
> |                              | Fine-tuning | Data Size                            | Model Size | Accuracy (%) |
> | ---------------------------------------- | ------------------ | ------------------------------------ | ---------- | ------------ |
> | Llama-3.1-Minitron-4B-Width              | FFT                | 20,000                               | 774M       | 77.00        |
> | Llama-3.1-Minitron-4B-Width              | LoRA               | 20,000                               | 774M       | 76.90        |
> | Llama-3.1-Minitron-4B-Width (FFT) ↪ Llama-3.1-70B | weak-to-strong | 20,000 (only trained on 4B-width) | 70B        | 78.67        |
> | Llama-3.1-Minitron-4B-Width (LoRA) ↪ Llama-3.1-70B | weak-to-strong | 20,000 (only trained on 4B-width) | 70B        | 78.52        |
>
> Based on this insight, our solution does not rely solely on fine-tuning but leverages the *weak-to-strong control mechanism* to redirect the large model’s powerful pretrained capabilities toward ToM tasks. Our method achieves consistent performance improvements, with accuracy gains of up to *1.67%* over direct fine-tuning. Notably, the weak-to-strong mechanism bridges the gap between smaller models (e.g., *4B-width*) and large-scale models (e.g., *70B*), demonstrating that the larger model’s inherent reasoning capacity can be effectively guided by the task-specific priors learned in smaller models. **This approach mitigates the difficulties posed by limited fine-tuning data and its mismatch and ineffectiveness to the intrinsic requirements of ToM tasks by effectively redirecting the larger model’s pretrained abilities to diverse ToM scenarios.**
>
> **References**:
>
> 1. *Parameter-Efficient Orthogonal Finetuning via Butterfly Factorization*, ICLR 2024.
> 2. *LoRA Learns Less and Forgets Less*, TMLR 2024 *(Featured Certification)*.
> 3. *Theory of Mind in Large Language Models: Examining Performance of 11 State-of-the-Art Models vs. Children Aged 7–10 on Advanced Tests*, Conference on Computational Natural Language Learning 2023.
> 4. *Boosting Theory-of-Mind Performance in Large Language Models via Prompting*, Johns Hopkins University, 2023. Retrieved from https://arxiv.org/abs/2304.11490.
>
> ---

---

> ### Author Response · Authors · 2024-11-25
>
> We hope that our response has helped explain our work's contributions. Please feel free to let us know if you have any further questions.

---

> > ### Comment · Reviewer_CTGe · 2024-12-03
> >
> > I thank the authors for their feedback and clarifications. Given these and the other reviews, I'll remain with my original assessment, again emphasizing the low confidence.

---

> > > ### Author Response · Authors · 2024-12-03
> > > **Thank you!**
> > >
> > > Thank Reviewer CTGe for their kind feedback and for maintaining the positive evaluation. We hope our discussions with all reviewers have helped better clarify the motivation and contributions of this study.

---

### Official Review · Reviewer_6PLd · 2024-11-04

**Soundness:** 3
**Presentation:** 2
**Contribution:** 2
**Rating:** 5
**Confidence:** 3

**Summary:**

The paper builds upon the MMToM-QA benchmark by introducing a method to align the general, amortized policy of a large LM (as originally used in MMToM-QA work) to the ToM tasks at hand using a post-trained expert small LM. The main contribution is a reweighting mechanism that integrates the extensive world knowledge from the large LLM with the ToM-specific policy from the small model. An extensive analysis on the improvement in performance as a result of this on the MMToM-QA benchmark. The paper thus effectively propose a way to improve upon the MMToM-QA method for better/more accurate Bayesian likelihood estimation of the goals and beliefs, that also now leverage the specialized knowledge corresponding to a particular ToM tasks in a scalable manner.

**Strengths:**

These are the strengths of the paper in my opinion:

1. Improves upon the Bayesian Inverse Planning Accelerated by Language Models, by making them more adaptive to ToM tasks at hand using a post-training process.

2. The proposed post-training method is more scalable and the integration of this expert post-trained knowledge to guide the large LM seems principled.

3. Extensive analysis and ablation studied on the MMToM-QA benchmark, effectively illustrating the methods adaptability and efficiency.

**Weaknesses:**

These are the weakness of the paper in my opinion:

1. The paper is very poorly written. Unless and until you go back and read the details from the appendix of MMToM-QA [1] method, the explanations are incomplete and hard to read. The paper would benefit from an appendix section and can be more upfront about the fact that this is a direct extension to the the MMToM-QA paper. For example, see the question 2, such implementation details are not discussed anywhere.

2. The paper is fairly incrimental as far as the methods used are concerned. The ToM inference is directly an extension of previous work, and so is the post-training method (eg: takes inspiration from the post-training literature, that uses a reweighting ratio between an expert and non-expert and subsequent normalizations [2][3]).

**Questions:**

1. Could you please explain why such a reweighting scheme between an expert and non-expert is a principled mechanism for policy adjustments/redirection in Equation 6 ? Is it related to some well-established theories in Bayesian / Probabilistic modelling literature (for example say Importance Sampling, or minimizing KL divergence between policies etc) ?? Expanding on this theoretical foundation would enhance the paper’s contribution and clarify the underlying principles.

2. How is the agent's belief distribution b(s) modelled? How is the belief evolution modelled $P\left(b_1^t \mid \hat{b}^{t-1}, s^t\right)$ ?? These details are missing. The paper would benefit from a thorough rewriting to make it more accessible to the reader.

3. How much time does this fine-tuning process take (specify the compute resources used etc) and what is the size of the pool $
\mathcal{D} = \left(s_i, b_i, g_i, a_i\right) _{i=1}^N$ in terms of N, used for optimizing eq 5?


References

1. MMToM-QA: Multimodal theory of mind question answering https://arxiv.org/pdf/2401.08743
2. DExperts: Decoding-time controlled text generation with experts and anti-experts. https://arxiv.org/abs/2105.03023
3. An Emulator for Fine-Tuning Large Language Models using Small Language Models. https://arxiv.org/pdf/2310.12962



--------
### **Post Discussion Update**

**I increase the score to a 5 for incorporating several suggestions, especially for a more accessible writing style post rebuttal. The effort of introducing the new theorem to justify the reweighting mechanism (under some strong initial assumptions) is also appreciated. Methodologically it's an incremental work extensively based on a recent paper MMToM-QA, but now without the need to explicitly post train the large model. Because of the limited novelty, I unfortunately cannot recommend a strong accept. However, I won't oppose if the other reviewers argue for an accept.**

---

> ### Author Response · Authors · 2024-11-22
> **Response to Reviewer 6PLd (1/4)**
>
> We sincerely thank Reviewer 6PLd for their valuable feedback and the opportunity to address the concerns. These perspectives provide important theoretical rationale and interdisciplinary context to our discussion. In response, we have incorporated Theorem 1 along with additional background information and have clarified the challenges associated with our approach in the revised manuscript to ensure a more thorough and enhanced understanding.
>
> ---
>
> **[W1&Q2: Writing and Clarity Issues; Need for Appendix and Implementation Details; Connection to MMToM-QA; Belief Distribution and Evolution Modelling]**
>
> We deeply appreciate your feedback, which has been valuable in improving the clarity and accessibility of the paper. In response to your concerns, we have made the following updates and clarifications:
>
> We agree on the importance of providing clearer explanations, especially regarding foundational concepts and their connection to implementation. To address this, **we have expanded the Introduction (lines 61-78) and largely rewritten the Methodology (pages 3 and 4) sections** to provide a more structured overview and motivation for our Bayesian framework, highlighting its extensions from MMToM-QA. These revisions emphasize the novel contributions of our scaled Bayesian reasoning framework while acknowledging the foundational role of MMToM-QA. Furthermore, we have added a new **Appendix B, summarizing MMToM-QA’s settings and our implementation details for reference.** Additionally, **the belief distribution and evolution modeling sections have been thoroughly rewritten** to integrate theoretical principles with practical implementation details.
>
> 1. **Question about modeling the belief distribution $b(s)$ and evolution $P(b^\tau \mid b^{\tau-1}, s^\tau)$:**
>
> **From a theoretical perspective**, the agent’s belief $b(s)$ is modeled as a probability distribution over possible object locations, simplifying the complex state space into manageable components. Belief evolution follows Bayesian updating principles:
> $P(b^\tau \mid b^{\tau-1}, s^\tau) = \frac{P(s^\tau \mid b^{\tau-1}) P(b^{\tau-1})}{P(s^\tau)},$
> where $P(s^\tau \mid b^{\tau-1})$ is likelihood of observing $s^\tau$ given the prior belief $b^{\tau-1}$, $P(b^{\tau-1})$ is prior belief from the previous step, and $P(s^\tau)$ is normalization term ensuring a valid probability distribution. For example, a belief $b(s)$ initially assigns equal probabilities to all potential locations of an object (e.g., *"the apple is in the basket or on the table"*). If an observation $s^\tau$ confirms the apple is in the basket, the belief is updated to increase the likelihood of the basket while reducing the likelihood of other locations.
>
> **In practice (for both our method and the MMToM-QA framework we follow), belief evolution is _not explicitly_ implemented as direct updates to $P(b^\tau \mid b^{\tau-1}, s^\tau)$;** _Instead_, the framework relies on approximating action likelihoods across different belief hypotheses $P(a^\tau \mid g, s^\tau, b^\tau)$. The language model processes symbolic prompts representing belief states and actions to compute these likelihoods, which are then compared to determine the most consistent belief. This approach **implicitly** updates the belief by evaluating how well it explains the observed actions.
>
> **_Implementation of implicit belief evolution in the pipeline_**
> ```python
> # Construct a symbolic prompt for belief evolution
> prompt = f"""
> Goal: {goal}
> State: {current_state}
> Belief: {current_belief}  # the belief is symbolically represented (e.g., as possible object locations [basket, table, ...]), as part of a prompt
> Action: {action}
> """
>
> # Tokenize input and compute prompt length
> inputs = tokenizer(prompt, return_tensors="pt", add_special_tokens=False)
> prompt_len = inputs['input_ids'].size(1)
>
> # Get log probabilities from the LM
> log_probs = language_model(**inputs).logits.log_softmax(dim=-1)
> action_tokens = tokenizer(action, return_tensors="pt")['input_ids']
>
> # Compute log probability of the action
> action_log_prob = log_probs.gather(index=action_tokens, dim=-1).sum()
>
> # Compare action log probabilities across beliefs (**implicit belief evolution**)
> final_prob = action_log_prob / action_tokens.size(1)
> ```
> This implementation does not directly compute  $P(b^\tau \mid b^{\tau-1}, s^\tau)$. Instead, it evaluates the action likelihoods  $P(a^\tau \mid g, s^\tau, b^\tau)$ for each belief hypothesis. The belief most consistent with observed actions is selected through probabilistic comparison. This demonstrates how symbolic prompts and LM outputs approximate the belief update $ P(b^\tau \mid b^{\tau-1}, s^\tau) $ without explicitly calculating each Bayesian term.
>
> **Initialization of $ P(b^0) $:**
> ```python
> # equal probabilities for all locations
> beliefs = torch.ones(num_locations) / num_locations
> ```
>
> This initialization reflects equal uncertainty across all possible locations, allowing subsequent observations to dynamically refine the belief.

---

> ### Author Response · Authors · 2024-11-22
> **Response to Reviewer 6PLd (2/4)**
>
> By integrating Bayesian principles with LM-based approximations, we bridge theory and practice to model belief evolution efficiently in complex multimodal scenarios. While formulas like:
> $P(b^\tau \mid b^{\tau-1}, s^\tau) = \frac{P(s^\tau \mid b^{\tau-1}) P(b^{\tau-1})}{P(s^\tau)}$
> remain foundational, symbolic prompts and LM outputs offer a scalable and efficient approximation. This approach builds on MMToM-QA while extending it to larger models and broader reasoning capabilities.
>
> The revised content aims to provide a comprehensive and accessible explanation of belief modeling, evolution, and their practical implementation. Thank you again for your constructive feedback.
>
> ---
>
> **[Q1: Could you please explain why such a reweighting scheme between an expert and non-expert is a principled mechanism for policy adjustments/redirection in Equation 6 ? Is it related to some well-established theories in Bayesian / Probabilistic modelling literature (for example say Importance Sampling, or minimizing KL divergence between policies etc)? Expanding on this theoretical foundation would enhance the paper’s contribution and clarify the underlying principles.]**
>
> We are happy to explain the principled mechanism underlying our weak-to-strong control framework for policy adjustment and redirection. While our weak-to-strong control empirically enables the large model to approximate the behavior of a directly tuned policy model ($\pi^*$) without direct parameter updates, we have also provided an updated theoretical rationale. These updates have been integrated into the revised manuscript on Pages 16–17, Appendix C.
>
> **Theorem:**
> Let $\pi^\mathcal{L}$ be a pretrained base model, $\pi^\mathcal{E}$ and $\pi^\mathcal{N}$ be smaller tunable models where $\pi^\mathcal{E}$ is fine-tuned on the target task, and $\pi^*$ be the directly tuned base model. Suppose the logit adjustment $\Delta s(X_t | x_{<t}) = s_{\pi^\mathcal{E}}(X_t | x_{<t}) - s_{\pi^\mathcal{N}}(X_t | x_{<t})$ approximates the scaled negative gradient of the cross-entropy loss for logits, i.e., $
> \Delta s \approx -\eta \nabla\_s \mathcal{L}\_{\text{CE}}(s\_{\pi^\mathcal{L}}, y),$
> where $\eta$ is the learning rate. Then, the proxy-tuned model $\tilde{\pi}$, defined by $s\_{\tilde{\pi}}(X\_t | x\_{<t}) = s\_{\pi^\mathcal{L}}(X\_t | x\_{<t}) + \Delta s(X\_t | x\_{<t}), $
> approximates the directly tuned base model $\pi^*$. The KL divergence between their output distributions is constrained by:
> $$
> D\_{\text{KL}}(P\_{\pi^*} \| P\_{\tilde{\pi}}) \leq \frac{\eta^2}{2} \lambda\_{\max} \|\nabla\_s \mathcal{L}\_{\text{CE}}(s\_{\pi^\mathcal{L}}, y)\|\_2^2 + \mathcal{O}(\eta^3),
> $$
> where $\lambda\_{\max}$ is the maximum eigenvalue of the Hessian of the CE loss for the logits.
>
>  **Proof:**
> When the learning rate $\eta$ is small, and the cross-entropy loss $\mathcal{L}_{\text{CE}}$ is smooth and twice differentiable with respect to the logits $s$, then the logit adjustment $\Delta s$ approximates the scaled negative gradient of the loss as:
> $$
> \Delta s \approx -\eta \nabla\_s \mathcal{L}\_{\text{CE}}(s\_{\pi^\mathcal{L}}, y).
> $$
>
> The logits of the directly tuned base model $\pi^*$ after fine-tuning are updated using gradient descent:
> $$
> s\_{\pi^*} = s\_{\pi^\mathcal{L}} - \eta \nabla_s \mathcal{L}\_{\text{CE}}(s\_{\pi^\mathcal{L}}, y) + \frac{\eta^2}{2} H\_s (\nabla_s \mathcal{L}\_{\text{CE}}(s\_{\pi^\mathcal{L}}, y)) + \mathcal{O}(\eta^3),
> $$
> where $H\_s$ is the Hessian of $\mathcal{L}\_{\text{CE}}$ with respect to the logits.
>
> The logits of the proxy-tuned model $\tilde{\pi}$ are:
> $$
> s\_{\tilde{\pi}} = s\_{\pi^\mathcal{L}} + \Delta s.
> $$
> When $\Delta s \approx -\eta \nabla\_s \mathcal{L}\_{\text{CE}}(s\_{\pi^\mathcal{L}}, y)$, we have:
> $$
> s\_{\tilde{\pi}} \approx s\_{\pi^\mathcal{L}} - \eta \nabla\_s \mathcal{L}\_{\text{CE}}(s\_{\pi^\mathcal{L}}, y).
> $$
> The difference in logits between the directly tuned model and the proxy-tuned model is:
> $$
> \epsilon_s = s_{\pi^*} - s_{\tilde{\pi}}.
> $$
> Then we consider their expressions:
> $$
> \epsilon\_s \approx \frac{\eta^2}{2} H\_s (\nabla\_s \mathcal{L}\_{\text{CE}}(s\_{\pi^\mathcal{L}}, y)) + \mathcal{O}(\eta^3).
> $$
> The KL divergence between the output distributions of $\pi^*$ and $\tilde{\pi}$ is constrained using the properties of the softmax function and the Lipschitz continuity of the KL divergence:
> $$
> D\_{\text{KL}}(P\_{\pi^*} \| P\_{\tilde{\pi}}) \leq \frac{1}{2} \|\epsilon\_s\|\_2^2.
> $$
> Considering the norm of $\epsilon\_s$:
> $$
> \|\epsilon\_s\|\_2^2 \approx \frac{\eta^4}{4} \|H\_s (\nabla\_s \mathcal{L}\_{\text{CE}}(s\_{\pi^\mathcal{L}}, y))\|\_2^2.
> $$
> The Hessian's norm is constrained by its maximum eigenvalue:
> $$
> \|H\_s (\nabla\_s \mathcal{L}\_{\text{CE}})\|\_2 \leq \lambda\_{\max} \|\nabla\_s \mathcal{L}\_{\text{CE}}(s\_{\pi^\mathcal{L}}, y)\|\_2,
> \\
> \|\epsilon\_s\|\_2^2 \leq \frac{\eta^4}{4} \lambda\_{\max}^2 \|\nabla\_s \mathcal{L}\_{\text{CE}}(s\_{\pi^\mathcal{L}}, y)\|\_2^2.
> $$

---

> ### Author Response · Authors · 2024-11-22
> **Response to Reviewer 6PLd (3/4)**
>
> The KL divergence is:
> $$
> D\_{\text{KL}}(P\_{\pi^*} \| P\_{\tilde{\pi}}) \leq \frac{\eta^2}{2} \lambda\_{\max} \|\nabla\_s \mathcal{L}\_{\text{CE}}(s\_{\pi^\mathcal{L}}, y)\|\_2^2 + \mathcal{O}(\eta^3).
> $$
> The KL divergence between the directly tuned model $\pi^*$ and the proxy-tuned model $\tilde{\pi}$ is constrained by terms involving the learning rate $\eta$, the gradient norm $\|\nabla\_s \mathcal{L}\_{\text{CE}}\|$, and the Hessian's maximum eigenvalue $\lambda\_{\max}$.
>
> In terms of **theoretical implications for practical applicability**, this analysis demonstrates that the weak-to-strong control mechanism **relies on the learned $\Delta s$ to approximate the scaled gradient $-\eta \nabla\_s \mathcal{L}\_{\text{CE}}(s\_{\pi^\mathcal{L}}, y)$ with higher-order terms contributing to the residual error:** importantly, our method does not require the small LM ($\pi^\mathcal{E}$) to strictly approximate the exact gradient of the cross-entropy loss for the large model. Instead, the large model ($\pi^\mathcal{L}$) leverages its **intrinsic capacity for generalization and adaptation**, based only on the approximate adjustment $\Delta s$ learned by the small LM.
>
> This inherent flexibility allows the large model to harness its **pre-trained potential**, activated by the weak-to-strong control mechanism, to effectively adapt to the current ToM task. Consequently, our method achieves **stable advanced performance** even in novel scenarios where the small LM provides only a coarse approximation of the gradient. This significantly reduces the reliance on strict fine-tuning and maximizes computational efficiency, ensuring the approach is both scalable and practical for the physical VirtualHome environment.
>
> ---
>
> **[W2: Incremental contribution; extension of previous work (post training, reweighting)]**
>
> We sincerely appreciate your feedback and the opportunity to address your concerns while clarifying the novelty of our work for a broader audience. While our solution surpasses the state-of-the-art solution by an impressive 4.6% representing a substantial improvement, we also discussed the theoretical foundation for its technical rationale (*here we are particularly grateful for your insightful feedback, which let us further explore and articulate this rationale through Theorem 1 and proof. Notably, few prior studies theoretically discuss why transferring fine-tuned behavior, such as ToM, from a smaller model to a larger one is effective*). The high-level principle derived from our theorem is that the smaller model can approximate the scaled gradient of the loss function for the larger model. This enables bypassing direct parameter updates in the large model while still capturing the primary adjustments necessary for fine-tuning. Crucially, the large model’s inherent capacity for generalization allows it to adapt effectively using approximate knowledge provided by the smaller model, reducing computational overhead and ensuring scalability.
>
> Guided by this *theoretical principle*, and stepping back to a broader perspective, prior studies have applied similar ideas to various domains (some of them are even awarded), including *toxicity avoidance* [1], *mitigating harmful output exposure* [2], *adjusting code generation and factuality* [3], *controlling sentiment* [4], and *reducing hallucinations or degeneration in text generation* [5, 6]. These works utilize reweighting mechanisms to approximate the behavior of language models as if directly fine-tuned, presenting a generalizable rationale across different contexts. When these studies take inspiration from this theoretical principle, they also provide new insights and applications in their respective domains, developing this research more inclusively. Additionally, some of these works [2, 4] have been recognized with outstanding paper awards, underscoring their broader impact and relevance within the *inclusive research community*.

---

> ### Author Response · Authors · 2024-11-22
> **Response to Reviewer 6PLd (4/4)**
>
> However, our scaled Bayesian ToM inference differs fundamentally. Rooted in a *cognitive-inspired Bayesian ToM framework* [7, 8, 9, 10], our method extends scalability from systemically developed cognitive science principles, specifically *inverse planning*, by introducing a scalable policy model (up to 70B and 405B parameters) for likelihood estimation in embodied simulators. Unlike prior techniques focusing on text generation tasks (e.g., code, sentiment, or toxicity control), our overall framework emerges from a different disciplinary background. It addresses unique challenges in *social cognition reasoning*. ToM tasks demand intricate reasoning and integration of world knowledge, making them distinctly more complex and multimodal in nature. Our method uses LMs to approximate large language model behaviors in these complex scenarios, enabling scalable and effective inference based on *Bayesian principles from cognitive science*.
>
> Finally, we believe our contributions enrich the *inclusive AI research community* in three key ways:
> 1. **New domain perspective**:  Address the scalability issues in the cognitive science-based framework by **scaling Bayesian Inverse Planning _up to 70B or even 405B sizes_** for social cognition reasoning, diverging fundamentally from text-generation tasks addressed in prior works.
> 2. **Theoretical rationale**: Establishing a scalable mechanism through *Theorem 1 and proof*, which allows smaller models to guide larger ones without direct parameter updates, thereby enabling efficient scalability.
> 3. **Engineering contributions**: Achieving ~4.6% improvement over existing best solutions, releasing open-source code, and introducing new datasets for training and testing in five unseen and diverse themes distinct from MMToM and other existing works.
>
>
> These contributions highlight the novelty of our work, demonstrating its scalability and adaptability across a wide range of ToM tasks. *We hope this clarifies the unique aspects and broader implications of our approach for a diverse audience, fostering inclusivity.* Thank you once again for your invaluable feedback, which has greatly enhanced our presentation and strengthened the manuscript.
>
> **References**:
>
> 1. *DExperts: Decoding-Time Controlled Text Generation with Experts and Anti-Experts*, ACL 2021.
> 2. *Emulated Disalignment: Safety Alignment for Large Language Models May Backfire!*, ACL 2024 *(Outstanding Paper Award)*.
> 3. *An Emulator for Fine-Tuning Large Language Models using Small Language Models*, ICLR 2024.
> 4. *Word Embeddings Are Steers for Language Models*, ACL 2024 *(Outstanding Paper Award)*.
> 5. *DoLa: Decoding by Contrasting Layers Improves Factuality in Large Language Models*, ICLR 2024.
> 6. *A Contrastive Framework for Neural Text Generation*, NeurIPS 2022.
> 7. *Action Understanding as Inverse Planning*, Cognition 2009.
> 8. *Building Machines that Learn and Think Like People*, Behavioral and Brain Sciences 2017.
> 9. *Goal Inference as Inverse Planning*, Proceedings of the Annual Meeting of the Cognitive Science Society 2007.
> 10. *Theory of Mind as Inverse Reinforcement Learning*, Current Opinion in Behavioral Sciences 2019.
>
> ---
>
> **[Q3: How much time does this fine-tuning process take (specify the compute resources used etc) and what is the size of the pool in terms of N, used for optimizing eq 5?]**
>
> The fine-tuning process for the smaller models (Llama3.1-8B) is carried out using a single NVIDIA H100 GPU. The training is conducted in BFoat16 mode to optimize memory usage, keeping the GPU memory consumption under 60GB. The process involves fine-tuning on the MMToM dataset's training split with the following configuration:
> - Batch size: 16 (achieved via a per-device batch size of 4 and gradient accumulation steps of 4),
> - Learning rate: $ 5 \times 10^{-5} $,
> - Number of epochs: 3.
> Under this setup, the fine-tuning process takes approximately **8 hours** to converge.
>
> The pool size $ N $ for optimizing Equation 5 is **20,000 data points**, sourced from our released data sampled from the embodied simulator and the MMToM-QA dataset. For training sets on transferred themes, the released training dataset also consists of **20,000 data points**, ensuring consistency.
>
> For the revised manuscript, we have included these details in our openly released code, Appendix C.2.1 and C.2.2, and explained the $N$ before Eq. (5). We hope that this information improves the clarity and accessibility of our approach.
>
> ---

---

### Official Review · Reviewer_W8eC · 2024-11-04

**Soundness:** 3
**Presentation:** 4
**Contribution:** 3
**Rating:** 6
**Confidence:** 4

**Summary:**

The paper presents a method for scaling Theory of Mind (ToM) capabilities in AI systems using a weak-to-strong Bayesian reasoning framework. The core approach combines Bayesian state inference with language models, where a smaller post-trained language model (4-8B parameters) guides the reasoning process of a larger model (up to 405B parameters) during test time1. The method works by transferring ToM-specific behaviors from the post-trained small language model to influence the latent reasoning of the larger language model. This approach enables the system to leverage both the extensive world knowledge of large language models and the ToM-specific behaviors learned through post-training, while avoiding the computational costs typically associated with post-training large models

**Strengths:**

- The weak-to-strong Bayesian framework provides a practical solution to scale Theory of Mind capabilities without post-training large models, addressing a key efficiency challenge in the field.
- The technical approach cleanly integrates Bayesian state inference with language models, providing a principled and effective foundation for ToM reasoning.
- The evaluation framework systematically tests both scalability and generalization, with clear ablations demonstrating the contribution of each component.
- The paper is well written, with a clear motivation, methodology, experimental section, and supporting figures.

**Weaknesses:**

- The paper doesn't adequately address the apparent contradiction: recent research shows smaller models have fundamental reasoning gaps, yet the method relies on a small model to guide larger model reasoning. This raises questions about whether the smaller model can effectively decompose problems it may struggle to reason about itself.

- The evaluation methodology could be strengthened by adopting a variance analysis framework similar to GSM-Symbolic (Mirzadeh et al., 2024, https://arxiv.org/pdf/2410.05229), which would help quantify how the model's Theory of Mind capabilities vary across different phrasings of the same underlying task.

**Questions:**

- Would you classify tom beliefs as a form of reasoning? If so, what makes a smaller model capable of classifying tom beliefs?
- Could the authors clarify if the primary mechanism of the weak-to-strong framework is to align the larger model's distribution with ToM-specific beliefs and task structure, while preserving its general reasoning capabilities, rather than directly transferring reasoning abilities from the smaller model? If so, this would suggest the smaller model acts more as a task-specific prior that guides the larger model's attention and beliefs, rather than as a direct source of reasoning. This interpretation could help explain the framework's effectiveness despite the typically limited reasoning capabilities of smaller models.
- Figure 1 is confusing, its not clear from the figure exactly how the smaller post trained lm conrols the larger lm. Why is this model denoted  'extreme'?
- How how is model performance impacted when the post trained lm is itself large (e.g. 70B class)?
- As stated in weakness 2, how does performance degrade when the task phrasing is edited?


Notes
- there are no line numbers

---

> ### Author Response · Authors · 2024-11-22
> **Response to Reviewer W8eC (1/8)**
>
> We would like to thank Reviewer W8eC for your thoughtful comments and questions regarding our submission. Here we address your concerns in the following.
>
> ---
>
> **[W1: The paper doesn't adequately address the apparent contradiction: recent research shows smaller models have fundamental reasoning gaps, yet the method relies on a small model to guide larger model reasoning. This raises questions about whether the smaller model can effectively decompose problems it may struggle to reason about itself.]**
>
> We appreciate the reviewer’s thoughtful feedback and the opportunity to clarify this point. The concern about smaller models’ inherent reasoning limitations in general contexts is valid; however, the proposed method addresses this issue through a targeted specialization approach. Smaller models are **not expected to independently solve complex reasoning tasks.** Instead, they are designed to encode ToM-specific behaviors, which represent likelihood inference patterns during Bayesian ToM reasoning, through a targeted post-training process. These specialized behaviors are then transferred to larger models during test time via the weak-to-strong control mechanism.
>
> This mechanism enables the smaller models to act as guides, directing the reasoning trajectories of the policy model (i.e., the larger model) to align with the structured requirements of Bayesian inverse planning. The Bayesian framework governs the entire inference process by defining posterior probabilities of mental states (beliefs and goals) based on observed actions and states. Through their specialized ToM-specific behaviors, the smaller models integrate into this framework, ensuring the larger models effectively leverage their extensive world knowledge and reasoning capabilities in ToM tasks.
>
> To address this explicitly, we revised the introduction (see Lines 61-78) to emphasize that the reasoning trajectory of the larger models is **not determined solely by the smaller models but is primarily structured by the Bayesian inverse planning framework, with the larger LM serving as the policy model to ensure world knowledge and implicit reasoning capabilities.**
>
> ---
>
> **[W2: The evaluation methodology could be strengthened by adopting a variance analysis framework similar to GSM-Symbolic (Mirzadeh et al., 2024, https://arxiv.org/pdf/2410.05229), which would help quantify how the model's Theory of Mind capabilities vary across different phrasings of the same underlying task.
> Q5: As stated in weakness 2, how does performance degrade when the task phrasing is edited?]**
>
> We appreciate the reviewer’s insightful suggestion and have learned GSM-Symbolic (Mirzadeh et al., 7 Oct 2024, https://arxiv.org/pdf/2410.05229). Their work highlights performance sensitivity in mathematical reasoning when task phrasing changes, such as altering proper names (e.g., people, foods, objects) or numbers, with accuracy variations of up to ±3.1% observed in certain cases. This suggests that the underlying reasoning processes in language models are often fragile and align more with in-distribution pattern matching than formal reasoning.
>
> Inspired by this, we conducted a variance analysis of our model's ToM capabilities across different thematic augmentations. Using the *Apartment* scenario as the in-distribution baseline and themes like *Andersen Fairy Tales*, *Ancient Egyptian*, and others as out-of-distribution (OOD) augmentations, we quantified the sensitivity of the model's ToM reasoning to such distribution shifts. The results are summarized in the Table A below:

---

> ### Author Response · Authors · 2024-11-22
> **Response to Reviewer W8eC (2/8)**
>
> **Table A: Variance Analysis of ToM Capabilities Across Thematic Augmentations**
> | Theme               | Scale              | 1.1  | 1.2  | 1.3  | Avg.  | 2.1   | 2.2   | 2.3   | 2.4   | Avg.  | All   |
> |---------------------|--------------------|-------|-------|-------|-------|-------|-------|-------|-------|-------|-------|
> |**Apartment (original)**| 70B-zero-shot      | 85.00 | 63.00 | 93.00 | 80.33 | 72.00 | 76.00 | 16.00 | 61.33 | 56.33 | 66.62 |
> |                     | 70B-post-train   | 91.00 | 69.00 | 95.00 | 85.00 | 69.33 | 80.00 | 29.33 | 69.33 | 62.00 | 71.86 |
> |                     | **8B ↪ 70B** | **90.00** | **74.00** | **93.00** | **85.67** | **74.67** | **77.33** | **70.67** | **76.00** | **74.67** | **79.38** |
> |**Andersen Fairy Tales**| 70B-zeroshot      | 88.00 | 73.00 | 90.00 | 83.67 | 70.67 | 80.00 | 25.33 | 66.67 | 60.67 | 70.52 |
> |                     | 70B-post-train     | 90.00 | 71.00 | 93.00 | 84.67 | 73.33 | 61.33 | 61.33 | 69.33 | 66.33 | 74.19 |
> |                     | 8B ↪ 70B           | 92.00 | 71.00 | 85.00 | 82.67 | 82.67 | 76.00 | 68.00 | 77.33 | 76.00 | 78.86 |
> | **Ancient Egyptian**| 70B-zeroshot      | 89.00 | 71.00 | 91.00 | 83.67 | 74.67 | 74.67 | 25.33 | 60.00 | 58.67 | 69.38 |
> |                     | 70B-post-train     | 89.00 | 69.00 | 96.00 | 84.67 | 72.00 | 76.00 | 61.33 | 64.00 | 68.33 | 75.33 |
> |                     | 8B ↪ 70B           | 90.00 | 73.00 | 88.00 | 83.67 | 69.33 | 76.00 | 73.33 | 74.67 | 73.33 | 77.76 |
> | **Outer Space**     | 70B-zeroshot      | 88.00 | 72.00 | 92.00 | 84.00 | 72.00 | 64.00 | 25.33 | 70.67 | 58.00 | 69.38 |
> |                     | 70B-post-train     | 91.00 | 68.00 | 90.00 | 83.00 | 69.33 | 65.33 | 61.33 | 68.00 | 66.00 | 75.33 |
> |                     | 8B ↪ 70B           | 90.00 | 70.00 | 92.00 | 84.00 | 73.33 | 81.33 | 66.67 | 78.67 | 75.00 | 77.76 |
> | **Wild West**       | 70B-zeroshot      | 88.00 | 72.00 | 92.00 | 84.00 | 72.00 | 64.00 | 25.33 | 70.67 | 58.00 | 69.14 |
> |                     | 70B-post-train     | 91.00 | 68.00 | 90.00 | 83.00 | 69.33 | 65.33 | 61.33 | 68.00 | 66.00 | 73.29 |
> |                     | 8B ↪ 70B           | 90.00 | 70.00 | 92.00 | 84.00 | 73.33 | 81.33 | 66.67 | 78.67 | 75.00 | 78.86 |
> | **Medieval Castle** | 70B-zeroshot      | 88.00 | 71.00 | 89.00 | 82.67 | 62.67 | 74.67 | 20.00 | 73.33 | 57.67 | 68.38 |
> |                     | 70B-post-train     | 85.00 | 69.00 | 89.00 | 81.00 | 65.33 | 69.33 | 57.33 | 68.00 | 65.00 | 71.86 |
> |                     | 8B ↪ 70B           | 90.00 | 72.00 | 89.00 | 83.67 | 72.00 | 76.00 | 68.00 | 84.00 | 75.00 | 78.71 |
>
> 1. **Theme-Level Variance:**
>    For **70B-zero-shot**, the performance varies significantly across themes, ranging from **66.62** in *Apartment* to **70.52** in *Andersen Fairy Tales* (variance of 3.90). In contrast, our solution **8B ↪ 70B** reduces this range to just **79.38** (Apartment) to **77.76** (*Ancient Egyptian*), with a variance of only 1.62. This demonstrates the ability of the weak-to-strong control mechanism to align the reasoning capabilities of larger LMs across varied thematic shifts.
>
> 2. **Task-Specific Variance (1.1 to 2.4):**
>    Within the *Andersen Fairy Tales* theme, we observe that: (i) counterfactual reasoning tasks (e.g., **2.3**, where beliefs are explicitly misaligned with reality) exhibit the most significant drops in performance for **70B-zero-shot** (25.33) compared to belief inference tasks requiring world knowledge (e.g., **1.1**, 88.00).
>    (ii) The variance between counterfactual reasoning (2.3) and world knowledge-based tasks (1.1) decreases with adaptation: **70B-post-train** reduces this variance from 62.67 to 8.00, while **8B ↪ 70B** achieves a minimal difference of 2.67.
>
> These findings align with GSM-Symbolic observations, where altering task phrasing (e.g., names or numbers) introduced performance degradation. Thematic shifts and task-specific requirements in ToM tasks introduce variance. Differently, our scalable Bayesian method, rather than overwriting the general pretraining capabilities of large LMs, leverage their foundational knowledge and reasoning capacities while guiding them through targeted post-training and weak-to-strong alignment. This approach ensures better stability to phrasing and distribution shifts, effectively redirecting the large LM’s general capabilities toward ToM-specific behaviors without compromising its broader generalization strengths.

---

> ### Author Response · Authors · 2024-11-22
> **Response to Reviewer W8eC (3/8)**
>
> To illustrate the thematic shifts and their potential impact on ToM reasoning, we summarize the primary changes in task phrasing across different scenarios. These transformations involve recontextualizing objects, environments, and interactions to reflect the distinct semantics and cultural nuances of each theme while preserving the underlying task structure:
>
> Table B: **Thematic Task Phrasing Variance Analysis**
> | Apartment (*original*)              | Andersen Fairy Tales   | Ancient Egyptian      | Wild West            | Outer Space           | Medieval Castle       |
> |---------------------|------------------------|-----------------------|----------------------|-----------------------|-----------------------|
> | apartment           | cottage               | palace                | saloon               | quarters              | saloon               |
> | bedroom             | chamber               | sleeping chamber      | bunk room            | sleeping quarters     | bunk room            |
> | bathroom            | washroom              | bathing room          | outhouse             | sanitation room       | outhouse             |
> | living room         | great hall            | audience hall         | bar area             | recreation area       | bar area             |
> | kitchen             | hearth                | kitchen               | cooking area         | replicator station    | cooking area         |
> | coffeetable         | wooden table          | stone table           | wooden table         | control console       | wooden table         |
> | desk                | writing desk          | writing table         | writing desk         | command station       | writing desk         |
> | kitchentable        | feasting table        | dining table          | dining table         | mess table            | dining table         |
> | sofa                | wooden bench          | cushioned bench       | wooden bench         | lounger               | wooden bench         |
> | kitchencabinet      | pantry                | storage chest         | storage shelf        | storage unit          | storage shelf        |
> | cabinet             | cupboard              | treasure chest        | supply cabinet       | storage unit          | supply cabinet       |
> | bathroomcabinet     | washstand             | washstand             | washstand            | hygiene compartment   | washstand            |
> | dishwasher          | washing basin         | servant               | wash basin           | sterilizer unit       | wash basin           |
> | fridge              | cooling box           | cool room             | icebox               | cold storage          | icebox               |
> | microwave           | heating stone         | heating pot           | stove                | food synthesizer      | stove                |
> | stove               | fireplace             | fire pit              | wood stove           | heating unit          | wood stove           |
> | apple               | apple                 | fruit                 | fresh apple          | synthesized apple     | fresh apple          |
> | book                | tome                  | papyrus scroll        | ledger               | data pad              | ledger               |
> | chips               | dried berries         | flatbread             | corn chips           | nutrition chips       | corn chips           |
> | condimentbottle     | spice jar             | spice jar             | sauce bottle         | flavor vial           | sauce bottle         |
> | cupcake             | honey cake            | honey pastry          | pastry               | synthesized pastry    | pastry               |
> | dishbowl            | clay bowl             | clay bowl             | ceramic bowl         | serving bowl          | ceramic bowl         |
> | plate               | wooden plate          | ceramic plate         | ceramic plate        | serving plate         | ceramic plate        |
> | remotecontrol       | magic wand            | scepter               | telegraph key        | control pad           | telegraph key        |
> | salmon              | smoked fish           | dried fish            | salted fish          | replicated fish       | salted fish          |
> | waterglass          | goblet                | chalice               | glass                | hydration vessel      | glass                |
> | wine                | mead                  | wine                  | whiskey              | synthesized wine      | whiskey              |
> | wineglass           | goblet                | goblet                | shot glass           | drinking vessel       | shot glass           |
>
> We have detailed these discussions, including the thematic task phrasing table and variance analysis, in Appendix D.5 and Figure 6.
>
> ---

---

> ### Author Response · Authors · 2024-11-22
> **Response to Reviewer W8eC (4/8)**
>
> ---
>
> **[Q1: Would you classify tom beliefs as a form of reasoning? If so, what makes a smaller model capable of classifying tom beliefs?]**
>
> We appreciate the reviewer’s thought-provoking question, which allows us to clarify the conceptual and methodological underpinnings of our approach. Below, we address the two parts of the query in turn.
>
> **Theory of Mind beliefs is a form of social reasoning:**  We classify Theory of Mind (ToM) beliefs as a specialized form of social reasoning, supported by prior findings that identify specific neurons as being responsible for complex social reasoning [4]. ToM involves reasoning about others' mental states—such as beliefs, intentions, and goals—based on observable actions and contextual cues. This process extends beyond mere pattern recognition to include inference, prediction, and social problem-solving, which are hallmarks of reasoning [1, 2]. For example, ToM reasoning allows individuals to predict future actions by inferring unobservable mental states, a capability critical for navigating dynamic social contexts [3, 4]. Unlike logical or mathematical reasoning, ToM reasoning is intersubjective, implicit in many cases, and context-dependent, making it unique among reasoning paradigms [5, 6]. Neuroscientific studies further support this perspective, showing that specific brain regions perform computations to represent and reason about others’ beliefs, reinforcing the idea that ToM involves cognitive processes akin to reasoning [7].
>
> **Smaller LMs' capabilities in classifying ToM beliefs are from large-scale pretraining:**  Smaller models, while limited in general reasoning capacity compared to larger models, can still effectively classify ToM beliefs through a combination of large-scale pretraining, alignment mechanisms, and targeted fine-tuning. The LLaMA-family models we use to benefit from extensive pretraining on diverse text corpora, enabling them to develop foundational linguistic and contextual understanding, including latent ToM-related capabilities [8, 9]. Alignment mechanisms, such as instruction tuning and Reinforcement Learning from Human Feedback (RLHF), further enhance these models' ability to handle social reasoning tasks. Instruction tuning rewards cooperative communication and context-sensitive responses, aligning model behavior with ToM-like reasoning, while RLHF improves alignment with human expectations, enhancing model performance on complex tasks [10, 11, 12, 13].
>
> In our approach, we further refine smaller models using targeted fine-tuning on the MMToM (Multimodal Theory of Mind) dataset, derived from simulated apartment environments in VirtualHome. This step enables the models to specialize in classifying ToM-specific behaviors, such as belief inference and goal prediction, within a controlled environment. Importantly, this fine-tuning complements the pretraining and alignment mechanisms, equipping smaller models with specialized capabilities without compromising their generalization potential [14]. Moreover, our weak-to-strong control mechanism transfers these specialized behaviors to larger models, ensuring that their broader reasoning capacities are effectively redirected toward ToM-specific tasks.
>
> Recent studies corroborate our findings. For instance, instruct-tuned LLMs have been shown to significantly outperform base LLMs on advanced ToM tasks, as instruction-tuning rewards cooperative principles fundamental to human communication and reasoning [10]. Similarly, RLHF has been demonstrated to enhance ToM task accuracy by guiding models toward socially aligned reasoning trajectories [12, 13]. Our findings align with these observations, showing that a combination of pretraining, alignment, and targeted fine-tuning equips smaller models with the capacity to classify ToM beliefs effectively.

---

> ### Author Response · Authors · 2024-11-22
> **Response to Reviewer W8eC (5/8)**
>
> **References**
> 1. *Theory of Mind*. *Internet Encyclopedia of Philosophy*. Retrieved from [https://iep.utm.edu/theomind/](https://iep.utm.edu/theomind/)
>
> 2. Baiano, C., Dolores, M., & Santangelo, G. (2024). Editorial: *Theory of Mind*. *Frontiers in Psychology, 15*, Article 1370048. [https://doi.org/10.3389/fpsyg.2024.1370048](https://doi.org/10.3389/fpsyg.2024.1370048)
>
> 3. *Theory of Mind*. *Neurobehavioral & Cognitive Disorders*. Retrieved from [https://www.medlink.com/articles/theory-of-mind](https://www.medlink.com/articles/theory-of-mind)
>
> 4. Jamali, M., Grannan, B. L., Fedorenko, E., Saxe, R., Báez-Mendoza, R., & Williams, Z. M. (2021). Single-neuronal predictions of others’ beliefs in humans. *Nature, 591*(7851), 610–614. [https://doi.org/10.1038/s41586-021-03260-8](https://doi.org/10.1038/s41586-021-03260-8)
>
> 5. *Theory of Mind*. Retrieved from *Wikipedia*: [https://en.wikipedia.org/wiki/Theory_of_mind](https://en.wikipedia.org/wiki/Theory_of_mind)
>
> 6. Van Duijn, M., van Dijk, B., Kouwenhoven, T., de Valk, W., Spruit, M., & van der Putten, P. (2023). *Theory of Mind in Large Language Models: Examining Performance of 11 State-of-the-Art Models vs. Children Aged 7–10 on Advanced Tests*. *Conference on Computational Natural Language Learning*. Retrieved from [https://aclanthology.org/2023.conll-1.25.pdf](https://aclanthology.org/2023.conll-1.25.pdf)
>
> 7. Moghaddam, S. R., & Honey, C. J. (2023). *Boosting Theory-of-Mind Performance in Large Language Models via Prompting*. *arXiv Preprint arXiv:2304.11490*. Retrieved from [https://arxiv.org/abs/2304.11490](https://arxiv.org/abs/2304.11490)
>
> 8. Gudibande, A., Wallace, E., Snell, C., Geng, X., Liu, H., Abbeel, P., Levine, S., & Song, D. (2023). *The False Promise of Imitating Proprietary LLMs*. Retrieved from [https://arxiv.org/abs/2304.12297](https://arxiv.org/abs/2304.12297)
>
> 9. Ye, J., Chen, X., Xu, N., Zu, C., Shao, Z., Liu, S., Cui, Y., Zhou, Z., Gong, C., Shen, Y., et al. (2023). *A Comprehensive Capability Analysis of GPT-3 and GPT-3.5 Series Models*. *arXiv Preprint arXiv:2303.10420*. Retrieved from [https://arxiv.org/abs/2303.10420](https://arxiv.org/abs/2303.10420)
>
> 10. Zhou, C., Liu, P., Xu, P., Iyer, S., Sun, J., Mao, Y., Ma, X., et al. (2024). *LIMA: Less Is More for Alignment*. *Advances in Neural Information Processing Systems, 36*.
>
> 11. Verhagen, A. (2015). Grammar and Cooperative Communication. In *Handbook of Cognitive Linguistics* (pp. 232–252).
>
> 12. Grice, P. (1975). Logic and Conversation. In *Syntax and Semantics, Vol. 3: Speech Acts* (pp. 41–58).
>
> 13. Tomasello, M. (2008). *Origins of Human Communication*. *MIT Press*.
>
> 14. Jin, C., Wu, Y., Cao, J., Xiang, J., Kuo, Y.-L., Hu, Z., Ullman, T., Torralba, A., Tenenbaum, J., & Shu, T. (2024). *MMToM-QA: Multimodal Theory of Mind Question Answering*. *Proceedings of the 62nd Annual Meeting of the Association for Computational Linguistics (ACL 2024)*.
>
> ---

---

> ### Author Response · Authors · 2024-11-22
> **Response to Reviewer W8eC (6/8)**
>
> ---
>
> **[Q2: Could the authors clarify if the primary mechanism of the weak-to-strong framework is to align the larger model's distribution with ToM-specific beliefs and task structure, while preserving its general reasoning capabilities, rather than directly transferring reasoning abilities from the smaller model? If so, this would suggest the smaller model acts more as a task-specific prior that guides the larger model's attention and beliefs, rather than as a direct source of reasoning. This interpretation could help explain the framework's effectiveness despite the typically limited reasoning capabilities of smaller models.]**
>
> Thank you for this insightful question. Yes, the weak-to-strong framework described in the paper indeed focuses on aligning the larger model's distribution with ToM-specific beliefs and task structures, while preserving its general reasoning capabilities, rather than primarily using the smaller model's reasoning abilities to estimate likelihood in the Bayesian inference. This interpretation aligns closely with our motivation and our experimental results.
>
> 1. The smaller model (e.g., 4B or 8B parameters) undergoes ToM-specific post-training, allowing it to learn task-relevant priors such as belief states and potential goals. These priors encode the structure of ToM tasks without requiring the smaller model to independently perform advanced reasoning. Instead, during inference, the smaller model acts as an assistive scaffold, conditioning the larger model’s likelihood estimation in a Bayesian framework. This role is encapsulated in the adjustment ratio: $\frac{\pi^{\mathcal{E}}}{\pi^{\mathcal{N}}},$
> where $\pi^{\mathcal{E}}$ is the post-trained smaller model’s task-specific policy, and $\pi^{\mathcal{N}}$ is the naive pre-trained smaller model’s policy.
>
> 1. During inference, the larger model (e.g., 70B or 405B parameters) integrates the adjustment ratio to refine its likelihood estimation dynamically. The overall policy is computed as:
> $\pi^{\mathcal{L}} \frac{\pi^{\mathcal{E}}}{\pi^{\mathcal{N}}},$, where $\pi^{\mathcal{L}}$ is the policy from the larger model, and the larger model retains its broad reasoning and world knowledge, ensuring that its capacity for generalization remains intact.
>
> 1. The framework establishes a clear division of labor between the smaller and larger models. The smaller model functions as a **ToM-specific lens**, providing structured guidance to refine the larger model’s predictions without limiting its extensive reasoning capacity. Meanwhile, the larger model leverages this guidance alongside its world knowledge and advanced reasoning abilities to tackle both straightforward and complex ToM tasks. This collaborative dynamic ensures that the framework capitalizes on the strengths of both models, enabling effective and efficient handling of diverse ToM scenarios.
>
> For example, as evidenced in the table below, when tested on five unseen themes, the **8B ↪ 70B model** consistently outperforms the **70B-post-trained model**, according to average results. The performance gains range from **+2.43** (e.g., in the "Ancient Egyptian" and "Outer Space" themes) to **+6.85** (e.g., in the "Medieval Castle" theme). These improvements demonstrate that the **8B ↪ 70B model** effectively preserves and transfers the larger model’s general reasoning capabilities among diverse unseen themes while aligning with ToM-specific requirements.

---

> ### Author Response · Authors · 2024-11-22
> **Response to Reviewer W8eC (7/8)**
>
> **Table A: Performance of the 8B ↪ 70B Model on Unseen Themes Compared to 70B-Post-Trained and 70B-Zero-Shot Models Across All ToM Tasks.**
>
> | Unseen Theme               | Scale              | 1.1  | 1.2  | 1.3  | Avg.  | 2.1   | 2.2   | 2.3   | 2.4   | Avg.  | All   |
> |---------------------|--------------------|-------|-------|-------|-------|-------|-------|-------|-------|-------|-------|
> |**Andersen Fairy Tales**| 70B-zeroshot      | 88.00 | 73.00 | 90.00 | 83.67 | 70.67 | 80.00 | 25.33 | 66.67 | 60.67 | 70.52 |
> |                     | 70B-post-train     | 90.00 | 71.00 | 93.00 | 84.67 | 73.33 | 61.33 | 61.33 | 69.33 | 66.33 | 74.19 |
> |                     | 8B ↪ 70B           | 92.00 | 71.00 | 85.00 | 82.67 | 82.67 | 76.00 | 68.00 | 77.33 | 76.00 | 78.86 |
> | **Ancient Egyptian**| 70B-zeroshot      | 89.00 | 71.00 | 91.00 | 83.67 | 74.67 | 74.67 | 25.33 | 60.00 | 58.67 | 69.38 |
> |                     | 70B-post-train     | 89.00 | 69.00 | 96.00 | 84.67 | 72.00 | 76.00 | 61.33 | 64.00 | 68.33 | 75.33 |
> |                     | 8B ↪ 70B           | 90.00 | 73.00 | 88.00 | 83.67 | 69.33 | 76.00 | 73.33 | 74.67 | 73.33 | 77.76 |
> | **Outer Space**     | 70B-zeroshot      | 88.00 | 72.00 | 92.00 | 84.00 | 72.00 | 64.00 | 25.33 | 70.67 | 58.00 | 69.38 |
> |                     | 70B-post-train     | 91.00 | 68.00 | 90.00 | 83.00 | 69.33 | 65.33 | 61.33 | 68.00 | 66.00 | 75.33 |
> |                     | 8B ↪ 70B           | 90.00 | 70.00 | 92.00 | 84.00 | 73.33 | 81.33 | 66.67 | 78.67 | 75.00 | 77.76 |
> | **Wild West**       | 70B-zeroshot      | 88.00 | 72.00 | 92.00 | 84.00 | 72.00 | 64.00 | 25.33 | 70.67 | 58.00 | 69.14 |
> |                     | 70B-post-train     | 91.00 | 68.00 | 90.00 | 83.00 | 69.33 | 65.33 | 61.33 | 68.00 | 66.00 | 73.29 |
> |                     | 8B ↪ 70B           | 90.00 | 70.00 | 92.00 | 84.00 | 73.33 | 81.33 | 66.67 | 78.67 | 75.00 | 78.86 |
> | **Medieval Castle** | 70B-zeroshot      | 88.00 | 71.00 | 89.00 | 82.67 | 62.67 | 74.67 | 20.00 | 73.33 | 57.67 | 68.38 |
> |                     | 70B-post-train     | 85.00 | 69.00 | 89.00 | 81.00 | 65.33 | 69.33 | 57.33 | 68.00 | 65.00 | 71.86 |
> |                     | 8B ↪ 70B           | 90.00 | 72.00 | 89.00 | 83.67 | 72.00 | 76.00 | 68.00 | 84.00 | 75.00 | 78.71 |
>
> Table B: Comparison of Overall Performance Gains of the 8B ↪ 70B Model Against the 70B-Post-Trained Model Across Unseen Themes.
>
> | **Unseen Theme**           | **Ours: 8B ↪ 70B** | **70B-post-train** | **Difference** |
> |---------------------|---------------|---------------------|----------------|
> | Andersen Fairy Tales | 78.86        | 74.19              | **+4.67**      |
> | Ancient Egyptian     | 77.76        | 75.33              | **+2.43**      |
> | Outer Space          | 77.76        | 75.33              | **+2.43**      |
> | Wild West            | 78.86        | 73.29              | **+5.57**      |
> | Medieval Castle      | 78.71        | 71.86              | **+6.85**      |
>
>
> We hope this clarifies the role of the weak-to-strong framework and the collaborative interplay between the smaller and larger models in the Bayesian ToM framework. **We also have incorporated these explanations into the revised manuscript (Appendix D.6) for enhanced clarity.**
>
> ---
>
> **[Q3: Figure 1 is confusing, it's not clear from the figure exactly how the smaller post-trained lm controls the larger lm. Why is this model denoted extreme?]**
>
> Thank you very much for your question and feedback. We have redrawn Figure 1 to improve clarity and focus. The revised diagram now has two key components:
> 1. The left block illustrates the overall framework of Bayesian ToM inference, showing how symbolic inputs are processed by the large LM while incorporating latent behavioral changes extracted from the smaller post-trained LM.
> 2. The right-hand block demonstrates the weak-to-strong mechanism in detail. It explains how the latent behaviour changes ($\Delta$) learned by the smaller post-trained LM are computed and subsequently applied to slightly adjust the likelihood estimates of the larger LM, effectively redirecting its output towards ToM-specific tasks.
>
> Regarding the term 'extreme', we have previously used it to refer to the significantly larger size of the LMs used in our method (e.g. 70B or even 405B parameters), which are much larger than the 7B/8B LMs used in previous work. However, for clarity and simplicity, we have removed the term "extreme" in the revised figure and now refer to it as the "large language model". This ensures consistency and avoids unnecessary ambiguity.
>
> In addition, to reduce visual clutter, the detailed data flow diagram from the previous version of Figure 1 has been moved to a supplementary figure in Appendix B. This adjustment allows the main figure to focus on high-level explanation, while still providing detailed insights for readers interested in the full data flow. We hope that this revision makes the figure easier to interpret and more in line with reviewers' expectations.
>
> ---

---

> ### Author Response · Authors · 2024-11-22
> **Response to Reviewer W8eC (8/8)**
>
> ---
> **[Q4: How how is model performance impacted when the post trained lm is itself large (e.g. 70B class)?]**
>
> Thank you for this question.  Table A provides the conducted experiments to analyze the impact of post-training when the language model itself is large (e.g., 70B-class models). The results reveal that direct post-training improves performance over the zero-shot baseline for both seen and unseen scenarios. For instance, on the **Apartment (seen)** scenario, the post-trained 70B model achieves an overall accuracy of **71.8%**, compared to **66.6%** for the zero-shot 70B baseline, reflecting a significant improvement. Similarly, for unseen scenarios like **Andersen Tales** and **Ancient Egyptian**, direct post-training results in average accuracy improvements of **+5.6% overall** (e.g., **74.1% vs. 70.2%** for Andersen Tales and **75.3% vs. 69.3%** for Ancient Egyptian). However, while these improvements demonstrate the benefits of post-training, further gains are observed with weak-to-strong pipelines that combine the task-specific priors from smaller models with the reasoning capacity of larger models.
>
> The **weak-to-strong pipelines** (e.g., 8B ↪ 70B or 8B ↪ 405B) achieve superior performance compared to directly post-trained 70B models, particularly in generalizing to unseen scenarios. On the **Apartment (seen)** scenario, the 8B ↪ 70B pipeline achieves an overall accuracy of **79.3%**, and the 8B ↪ 405B pipeline achieves **81.3%**, significantly outperforming the directly post-trained 70B model's accuracy of **71.8%**. Across unseen environments, the 8B ↪ 70B pipeline achieves an average accuracy of **78.7%**, outperforming the directly post-trained 70B model’s **73.2%** by **+5.5%**. The 8B ↪ 405B pipeline further improves performance, with an average accuracy of **80.2%** across unseen environments. These results highlight the benefits of weak-to-strong approaches, which leverage task-specific priors from smaller models to guide larger models effectively, enhancing both generalization and overall accuracy.
>
> As evidenced, while post-trained large models (e.g., 70B) improve over zero-shot baselines, the weak-to-strong pipelines demonstrate better performance across both belief inference and goal inference tasks. This approach optimally utilizes the complementary strengths of smaller and larger models, significantly enhancing accuracy in unseen environments. We have incorporated these insights and numerical evidence into the revised manuscript for clarity.
>
> Table A: Transfer performance of the Bayesian method with different scaling settings (zero-shot, direct post-training, and weak-to-strong control) from the Apartment scenario to various unseen environments. All models are based on Llama 3.1.
>
> | **Solution**        | **Apartment (seen)** | **Andersen Tales** | **Ancient Egyptian** | **Outer Space** | **Wild West** | **Medieval Castle** |
> |---------------------|----------------------|---------------------|-----------------------|-----------------|--------------|---------------------|
> | **Raw**             |                      |                     |                       |                 |              |                     |
> | 70B-zero-shot       | 80.3 / 56.3 / 66.6  | 83.6 / 60.6 / 70.2  | 83.6 / 60.6 / 69.3    | 84.0 / 58.0 / 69.1 | 82.6 / 57.6 / 68.3 | 82.6 / 57.6 / 68.3  |
> | 70B-post-trained    | 85.0 / 62.0 / 71.8  | 84.6 / 66.3 / 74.1  | 84.6 / 66.3 / 75.3    | 83.0 / 66.0 / 73.2 | 81.0 / 65.0 / 71.8 | 81.0 / 65.0 / 71.8  |
> | **Ours**            |                      |                     |                       |                 |              |                     |
> | 4B-wide ↪ 70B       | 83.6 / 74.6 / 78.5  | 84.0 / 75.3 / 79.0  | 83.0 / 75.3 / 79.1    | 82.6 / 75.3 / 78.4 | 84.0 / 74.6 / 78.6 | 84.6 / 73.0 / 78.0  |
> | 4B-depth ↪ 70B      | 84.6 / 73.6 / 78.3  | 85.0 / 71.3 / 77.1  | 85.3 / 71.3 / 77.9    | 81.6 / 71.0 / 75.5 | 83.3 / 71.3 / 76.4 | 83.3 / 64.0 / 72.2  |
> | 8B ↪ 70B            | 85.6 / 74.6 / 79.3  | 82.6 / 76.0 / 77.8  | 83.6 / 76.0 / 77.7    | 84.0 / 75.0 / 78.8 | 83.3 / 74.0 / 78.0 | 83.6 / 75.0 / 78.7  |
> | 8B ↪ 405B           | 87.0 / 77.0 / 81.3  | 85.8 / 76.0 / 80.2  | 86.0 / 76.3 / 80.4    | 87.2 / 75.5 / 80.5 | 85.3 / 76.0 / 79.9 | 85.6 / 75.2 / 79.7  |
>
> ---
>
> **[Notes: there are no line numbers]**
>
> Thank you for this kind note. We have now included line numbers in the revised manuscript for easier navigation.
>
> ---

---

### Official Review · Reviewer_dq3d · 2024-11-06

**Soundness:** 2
**Presentation:** 2
**Contribution:** 2
**Rating:** 6
**Confidence:** 2

**Summary:**

This paper presents a scalable approach to enhance machine-based ToM, which is crucial for understanding and predicting human-like mental states in multimodal environments. The authors propose a weak-to-strong Bayesian reasoning framework that leverages the strengths of large language models to improve ToM inference without the computational costs associated with extensive post-training.

**Strengths:**

1. The author's approach seems to be novel
2. I like the experimentation, did show the scaling trend.

**Weaknesses:**

1. Presentation could be better. For instance, Fig 1, it contains too much information.
2. The major concern here is that everything is done on Lora. What happen if you do full-scale training? How much performance improvement do you expect to gain?
3. Didn't test smaller model such as phi-3b, gemma-2b etc. Maybe you can run full fine-tuning on smaller model.

**Questions:**

1. My major question is, how much improvement do you expect to gain if you run full-scale training?
2. Would be nice to see if you can test this on a smaller model, GPT2, or gemma-2b and see if there is larger performance gain.
3. The performance difference between llama2 vs llama3 vs llama3.1 isn't that much, is it because pre-training (or the quality of the pre-trained model) matter less in this context? Do you have theory/explaination why this happens?

---

> ### Author Response · Authors · 2024-11-22
> **Response to Reviewer dq3d (1/3)**
>
> We sincerely thank Reviewer dq3d for their thoughtful acknowledgment and affirmation of our philosophy on leveraging scaling. In response to your comments, we have provided detailed answers below and integrated them into our revised manuscript to enhance clarity and comprehensiveness.
>
> ---
>
> **[W1: Presentation could be better. For instance, Fig 1, it contains too much information.]**
>
> We implement several changes to enhance readability and clarity:
> 1. We have redrawn Figure 1 to improve clarity and focus. The revised figure now consists of two key components: The left block illustrates the overall framework of Bayesian ToM inference, showing how symbolic inputs are processed through the large LM while incorporating latent behavior changes extracted from the smaller post-trained LM. The right block demonstrates the weak-to-strong mechanism in detail. It explains how the latent behavior changes ($\Delta$) learned by the post-trained smaller LM are computed and subsequently applied to refine the likelihood estimates of the larger LM, effectively redirecting its output toward ToM-specific tasks.
>
> 2. To explain the methodology in Figure 2 better, the methodology section (pages 3 and 4) has been restructured and massively rewritten to improve accessibility and clarity, highlighted in colored text.
>
> ---

---

> ### Author Response · Authors · 2024-11-22
> **Response to Reviewer dq3d (2/3)**
>
> ---
> **[W2&W3&Q1&Q2: The experiments primarily rely on LoRA fine-tuning. What would happen if full-scale fine-tuning (FFT) was applied instead? Could it result in significant performance improvements? Additionally, have smaller models, such as Phi-3B, Gemma-2B, or GPT2, been evaluated using FFT to explore potential performance gains?]**
>
>
> We appreciate your insightful and constructive questions because they provided us with the opportunity to explore three crucial aspects:
> 1. The effectiveness of LoRA compared to full-scale training (FFT) in capturing Theory of Mind (ToM) behaviors, and
> 2. The estimated performance improvements achievable through FFT from a scaling perspective, particularly for smaller models.
> 3. The stability of our weak-to-strong control mechanism across different fine-tuning methods and model scales:
>
> To address these points, we conducted additional experiments, including comparisons between LoRA and FFT across different data sizes and fine-tuning on smaller models. The experiments were performed using 20,000 data points, over two epochs, on 8 NVIDIA A100 80GB GPUs. The results are summarized in Tables A and B below:
>
> **Table A: Comparison of direct full-finetuning training (FFT) and LoRA fine-tuning for GPT2-large and Gemma-2B across different data sizes.**
> | Model Name  | Fine-tuning Method | Data Size | Model Size | Accuracy (%) |
> | ----------- | ------------------ | --------- | ---------- | ------------ |
> | GPT2-large  | FFT                | 20,000    | 774M       | 63.4         |
> | GPT2-large  | LoRA               | 20,000    | 774M       | 62.4         |
> | GPT2-large  | FFT                | 8,000     | 774M       | 62.8         |
> | GPT2-large  | LoRA               | 8,000     | 774M       | 62.1         |
> | gemma-2b    | FFT                | 20,000    | 2B         | 68.8         |
> | gemma-2b    | LoRA               | 20,000    | 2B         | 68.5         |
> | gemma-2b    | FFT                | 8,000     | 2B         | 68.2         |
> | gemma-2b    | LoRA               | 8,000     | 2B         | 67.9         |
>
>
> **Table B: Comparison of Llama-3.1-70B controlled by direct full-finetuning training (FFT) and LoRA fine-tuning for Llama-3.1-Minitron-4B-Width via our weak-to-strong control mechanism.**
> | Model Name                               | Fine-tuning Method | Data Size                            | Model Size | Accuracy (%) |
> | ---------------------------------------- | ------------------ | ------------------------------------ | ---------- | ------------ |
> | Llama-3.1-Minitron-4B-Width              | FFT                | 20,000                               | 774M       | 77.00        |
> | Llama-3.1-Minitron-4B-Width              | LoRA               | 20,000                               | 774M       | 76.90        |
> | Llama-3.1-Minitron-4B-Width (FFT) ↪ Llama-3.1-70B | weak-to-strong control | 20,000 (only trained on 4B-width) | 70B        | 78.67        |
> | Llama-3.1-Minitron-4B-Width (LoRA) ↪ Llama-3.1-70B | weak-to-strong control | 20,000 (only trained on 4B-width) | 70B        | 78.52        |
>
> The results show the following trends:
> 1. When sufficient training data (e.g. 20,000 data points) is available, full-scale training (FFT) outperforms LoRA by 0.9-1.2 percentage points, indicating the effectiveness of FFT in exploiting richer data.
> 2. The performance gap between FFT and LoRA increases with larger training datasets (e.g. 20,000 vs. 8,000 data points), highlighting that FFT benefits more from increased data compared to LoRA.
> 3. The gap between FFT and LoRA decreases as model size increases (e.g. Gemma-2B vs. GPT2-large-774M). Larger models appear to adapt better to LoRA fine-tuning, thus narrowing the performance gap.
> 4. Using our weak-to-strong control mechanism, Llama-3.1-Minitron-4B-Width (trained via FFT or LoRA) transferred its knowledge effectively to Llama-3.1-70B, achieving 78.67% and 78.52% accuracy, respectively. The results demonstrate that our approach is largely insensitive to the choice of fine-tuning method under popular LMs' size, indicating that the mechanism can leverage ToM-specific behaviors elicited by both FFT and LoRA with high efficiency.
>
> Our experiments demonstrate that FFT can achieve modest performance improvements over LoRA, especially with sufficient data. However, the performance gap is minimal for larger models, making LoRA an efficient and less resource-intensive alternative. Importantly, our weak-to-strong control mechanism shows robust performance and is less sensitive to the fine-tuning method, enabling scalable and effective ToM-specific behavior elicitation in larger models.
>
> These findings and detailed analyses have been included in the revised manuscript (see Appendix B.3) for a comprehensive comparison. Thank you for your constructive feedback, which has greatly enriched our evaluation.
>
> ---

---

> ### Author Response · Authors · 2024-11-22
> **Response to Reviewer dq3d (3/3)**
>
> ---
>
> **[Q3: The performance difference between llama2 vs llama3 vs llama3.1 isn't that much, is it because pre-training (or the quality of the pre-trained model) matters less in this context? Do you have theory/explaination why this happens?]**
>
> Thanks for this insightful question. Based on the experimental results in Table 2 and Table 3, the impact of pre-training (or the quality of the pre-trained model) diminishes primarily due to a **ceiling effect**, but only when comparing large LMs (e.g. within the 70B parameter range). In contrast, the importance of pre-training becomes more apparent when comparing smaller and larger models. For example, in Llama-2, moving from 7B to 70B after ToM-specific post-training leads to a 6% improvement in belief inference accuracy (from 76.33% to 82.33%) and a 2% improvement in goal inference accuracy (from 70% to 72%). This highlights the role of scale in encoding richer representations.
>
> When we explore the finer reasons why the impact of pre-training becomes less significant at larger scales (e.g. the comparison in Table 2 of Llama-2-**70B** pre-trained with 2.2 trillion tokens vs. Llama-3.1-**70B** pre-trained with 15 trillion tokens), we find that Llama-3.1 benefits substantially from larger pre-training corpora, which improve performance primarily on tasks that rely more on world knowledge compared to those that require long-term logical reasoning:
>
> 1. **Belief inference tasks (world knowledge and short-term reasoning):**.
>    Belief inference tasks rely heavily on **world knowledge** and basic **short-term reasoning**. For example, recognising true or false beliefs often requires knowledge of typical human behaviour or object interactions, which larger pre-training corpora help to refine. Here, Llama-3.1 achieves a 3.67% improvement (from 83.00% to 85.67%) over Llama-2, indicating that scaling pre-training data improves the representation of world knowledge.
>
> 2. **Goal inference tasks (long-term reasoning):**.
>    Goal inference tasks require **long-term reasoning**, such as integrating temporal observations, dynamically updating beliefs, and implicitly reasoning about counterfactuals where actions are guided by incorrect or evolving beliefs. Larger pre-training corpora can marginally improve both logical and counter-reasoning, but these tasks are often more dependent on the **fine-tuning stage** and the reasoning framework employed (e.g. weak-to-strong control). For example, counter-reasoning may involve predicting an agent's goal when its belief does not match reality, requiring nuanced updates based on hypothetical alternatives. For goal inference, the improvement from Llama-2 to Llama-3.1 is smaller, at 1.67% (from 72.33% to 74.00%). This suggests that, beyond a certain scale, pre-training quality alone has a limited impact on tasks that emphasize complex logical or counter-reasoning, as the reasoning bottleneck shifts to task-specific fine-tuning.
>
> In summary, pre-training quality plays a critical role in **smaller-scale models** and tasks that rely heavily on **world knowledge** (e.g., belief inference). However, for **larger scale models** (e.g. 70B parameters), its influence diminishes due to: **(i) ceiling effects**: Pre-training saturates its contribution when basic representations are already robust. **(ii) Task-specific bottlenecks**: Logical reasoning tasks (e.g. goal inference) depend more on task-specific fine-tuning than on incremental pre-training improvements.
>
> We have included these discussions in our revised manuscript (see Appendix C.4) to provide enhanced clarity on the interaction between pre-training, model scale, and task complexity in ToM reasoning benchmarks.
>
> ---

---

> ### Author Response · Authors · 2024-11-25
>
> We hope that our response has helped explain our work's contributions. Please feel free to let us know if you have any further questions.

---

### Meta-Review · Area_Chair_5VJ3 · 2024-12-24

**Metareview:**

This paper proposes a method to scale theory of mind through weak-to-strong generalization. TOM is important to understand how and predict human-like states. Most reviewers agree that the proposed method is scalable, and with the new introduction of theorem, it justifies the alignment process without the explicit post training process. However, three main concerns. Firstly, presentation, it is hard to follow and also because of this, some of “could-be” important contributions could not be fully appreciated. Secondly, the novelty of the proposed method. The proposed method is largely an extension of the prior work. Third, improvements need more justifications in the experiments. This paper is an interesting one, yet it does require a lot of polishing to be ready, in terms of both better presentation to showcase its novelty and contribution and secondly more systematic experiments (even on more datasets) to justify its benefits (though this point is improved in the rebuttal).

**Additional Comments On Reviewer Discussion:**

Reviewers almost all raised concerns about the presentation clarity, which still remains an issue after the rebuttal. In terms of this paper's contribution, the newly added theorem gives it more justification, and one reviewer raised one score because of it. However, due to its presentation unclarity, It does need more polishing to better showcase its added novel contribution.

---

### Decision · Program_Chairs · 2025-01-22

Reject